# ZEUS: Zero-shot Embeddings
# for Unsupervised Separation of Tabular Data

**Patryk Marszałek**[1,2]    **Tomasz Kuśmierczyk**[*,1]    **Witold Wydmański**[1,2]    **Jacek Tabor**[1]

**Marek Śmieja**[*,1]

[1]Faculty of Mathematics and Computer Science, Jagiellonian University, Kraków, Poland
[2]Doctoral School of Exact and Natural Sciences, Jagiellonian University, Kraków, Poland

{t.kusmierczyk,witold.wydmanski,jacek.tabor,marek.smieja}@uj.edu.pl
patryk.marszalek@doctoral.uj.edu.pl

## Abstract

Clustering tabular data remains a significant open challenge in data analysis and machine learning. Unlike for image data, similarity between tabular records often varies across datasets, making the definition of clusters highly dataset-dependent. Furthermore, the absence of supervised signals complicates hyperparameter tuning in deep learning clustering methods, frequently resulting in unstable performance. To address these issues and reduce the need for per-dataset tuning, we adopt an emerging approach in deep learning: zero-shot learning. We propose ZEUS, a self-contained model capable of clustering new datasets without any additional training or fine-tuning. It operates by decomposing complex datasets into meaningful components that can then be clustered effectively. Thanks to pre-training on synthetic datasets generated from a latent-variable prior, it generalizes across various datasets without requiring user intervention. To the best of our knowledge, ZEUS is the first zero-shot method capable of generating embeddings for tabular data in a fully unsupervised manner. Experimental results demonstrate that it performs on par with or better than traditional clustering algorithms and recent deep learning-based methods, while being significantly faster and more user-friendly.

## 1 Introduction

Clustering remains a fundamental yet challenging task in unsupervised learning. It is particularly hard for tabular data, which inherently lacks the structured spatial or semantic properties of images or texts. Unlike for image clustering, where intrinsic visual similarities can guide cluster formation, defining meaningful similarities in tabular data is highly dataset-specific, complicating the generalization of clustering methods across diverse applications. Recent developments leveraging deep learning have demonstrated promise in generating richer representations for clustering tasks. However, these methods frequently suffer from instability due to their sensitivity to hyperparameter selection, a challenge exacerbated by the absence of supervised signals to guide optimization. Consequently, practitioners working with tabular data often resort to simpler, classical algorithms like k-means [23], despite their limited capacity for capturing complex underlying data structures, simply to avoid extensive manual tuning.

---

[*]Joint contribution to project conception, design, and research supervision.

39th Conference on Neural Information Processing Systems (NeurIPS 2025).

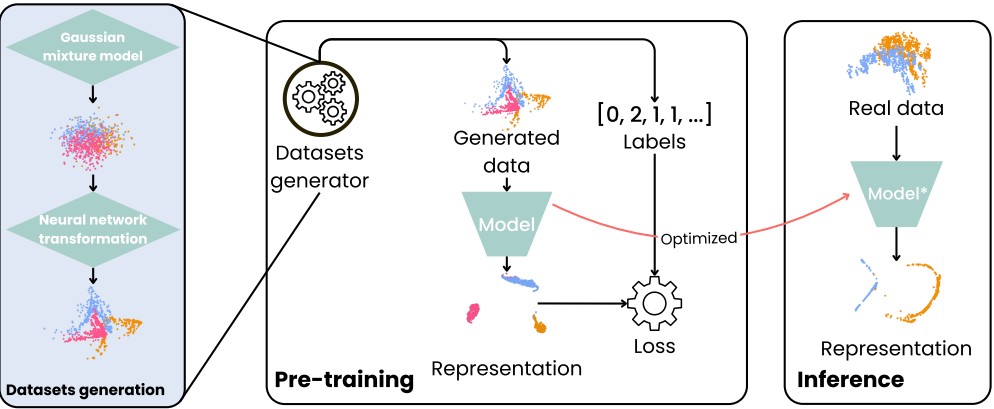

Figure 1: Schematic characterization of ZEUS: (left) synthetic datasets generation; (middle) pre-training on datasets with known labels; (right) deployment of a frozen model for real-world tasks.

We address these challenges with ZEUS – a **z**ero-shot transformer-based model for **e**mbedding new tabular datasets in a form convenient for **u**nsupervised **s**eparation (=clustering), without the need for additional fine-tuning. Given a new dataset, ZEUS returns its transformed representation, where clusters can be easily discovered using simple methods, like k-means. Since it works as a zero-shot learner, it significantly reduces hyperparameter tuning complexity and computation time, enabling effective clustering in seconds. It is a plug-and-play solution that requires no fine-tuning and runs in a single forward pass for new datasets.

Inspiration for ZEUS stems from the Prior-data Fitted Networks (PFNs) [16], a recently introduced framework highlighting the potential of in-context learning for tabular data. PFNs operate by feeding a transformer [31] contexts representing complete tasks, i.e., training data along with query samples for which the model predicts labels. Despite their impressive performance, they are limited to supervised problems. ZEUS extends this basic idea for unsupervised tasks by addressing the two key challenges of how to: (1) generate prior synthetic data with clear but non-trivial clustering structures for pre-training; and (2) encode prior clustering knowledge for new unlabeled datasets. Additionally, in contrast to TabPFN, ZEUS is a zero-shot model and during inference does not rely on any context labels.

Unlike traditional methods that optimize arbitrary heuristics (e.g., DEC [32]), ZEUS approaches clustering by learning how to invert data generation processes. In particular, it is pre-trained on synthetic datasets to learn how to infer cluster assignments. The datasets have known latent structures, which enables supervised guidance for the training, and by generating diverse datasets we supply it with the prior knowledge about what can constitute a possible clustering structure, enabling effective generalization to new real-world datasets during inference. Figure 1 illustrates the key concepts of the method while Figure 2 presents representations generated by ZEUS.

Experiments demonstrate that ZEUS consistently matches or surpasses both classical clustering algorithms and recent deep learning-based approaches, establishing it as a powerful and practical tool for the unsupervised analysis of tabular data. Beyond its strong empirical performance, we also provide theoretical justification by showing that ZEUS fits the framework of Prior-data Fitted Networks, thereby reinforcing its theoretical soundness (see Section 2.3).

We supplement the paper with an appendix that includes background on prior-data fitted networks, specifics of the synthetic data generation process, details of the experimental results, and additional experimental studies. The code used in this paper is available at https://github.com/gmum/zeus.

## 2 Pre-training representation for clustering

In this section, we introduce our approach to zero-shot representation pre-training for clustering. First, we characterize the proposed probabilistic model and its training objective. Then, we explain the process of generating synthetic datasets for pre-training. Finally, we highlight the connection between our model, Bayesian learning and the framework for prior-data fitted networks.

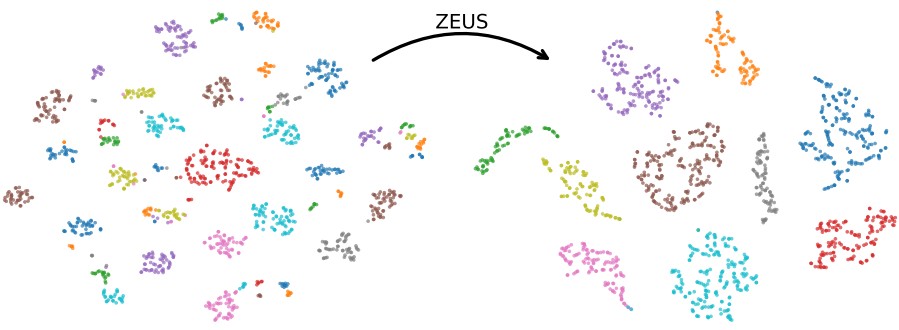

Figure 2: t-SNE visualization for a sample synthetic dataset. The representation from ZEUS (right panel) significantly improves consistency with ground-truth classes and reveals a clearer data structure.

## 2.1 ZEUS method

**Zero-shot pre-training.** Traditional machine learning models $f_\theta$ are trained by optimizing parameters $\theta$ to solve a specific *fixed* task, such as classification or clustering, over a fixed dataset. On the other hand, *in-context learning* (ICL) [9, 21, 14] enables *pre-trained models* $f_{\theta^*}$ to adapt to new tasks or datasets without ever updating the pre-trained parameters $\theta^*$. Similarly, *zero-shot learning* allows models to address problems they were never explicitly trained on, without requiring any examples of the target task. Unlike ICL, which adapts through demonstration examples, zero-shot approaches rely entirely on knowledge encoded during pre-training. The pre-training of zero-shot models typically involves massive and diverse datasets that encourage the model to learn rich, generalizable representations of underlying patterns and relationships. This is usually done using *transformers*, as they both can process set-valued inputs (e.g., accept a whole dataset at once) and model complex dependencies.

Our approach involves pre-training a transformer $f_\theta$ on a diverse collection of datasets $\mathcal{D}$ such that $\mathcal{D} := (\mathbf{x}, \mathbf{y})$, where $\mathbf{y} = \{y_i\}$ denotes ground-truth labels (e.g., clusters) and $\mathbf{x} = \{x_i\}$ are feature vectors. A loss $\mathcal{L}$ encourages $f_\theta$ to assign accurate labels $\mathbf{y}$ to all data points in the target datasets $\mathcal{D}$:

$$\theta^* = \arg\min_\theta \, \mathbb{E}_{p(\mathcal{D})}\big[\mathcal{L}\big(\mathbf{y}, \, f_\theta(\mathbf{x})\big)\big] \approx \arg\min_\theta \, \frac{1}{N_\mathcal{D}} \sum_{\mathcal{D} \sim p(\mathcal{D})} \mathcal{L}\big(\mathbf{y}, \, f_\theta(\mathbf{x})\big), \tag{1}$$

where $p(\mathcal{D})$ is a process generating *prior datasets* which *implicitly* characterize a space of possible target tasks (here approximated with $N_\mathcal{D}$ Monte-Carlo samples). Note, the loss $\mathcal{L}$ *may factorize* over individual points as $\mathcal{L}(\mathbf{y}, \hat{\mathbf{p}}) = \sum_i \ell(y_i, \hat{p}_i)$, where $\hat{p}_i$ denotes a probabilistic prediction for an individual input $x_i$, but unlike for classification, for tasks such as clustering or anomaly detection, the predictions $\hat{p}$ must be made *jointly* for all inputs. Hence, $f_\theta(\mathbf{x})$ can not be decomposed into predictions for individual inputs $x_i$. Furthermore, in contrast to typical supervised ICL, training labels are not included in the context, as they are unavailable during inference. This makes the unsupervised learning problem significantly more challenging, since we cannot explicitly guide the model toward the desired structure in a given dataset.

**Representation learning for probabilistic clustering.** We pre-train an encoder $f_\theta$ to map inputs $\mathbf{x}$ to representation vectors $z(x_i) = f_\theta(\mathbf{x})_i \in \mathbb{R}^D =: S$. We aim at positioning data points in $S$, so that they naturally form probabilistic clusters, and our goal is to make the encoder *find better representations for target tasks*. This objective we frame as maximizing the log-likelihood of correct cluster assignments. Since the ground-truth cluster assignments $y_i$ are available during pre-training, this is expressed as:

$$\mathcal{L}_{prob} = -\sum_i \log p_{y_i}(x_i). \tag{2}$$

For specifying $p_y(x)$, we draw inspiration from Gaussian Mixture Models (GMMs) [7, 15], where each cluster $k$ is represented by a Gaussian distribution centered at $c_k$, and data points are probabilistically assigned to clusters through probabilities $p_k(x_i)$, based on their proximity to these centroids. Consequently, alongside the encoded representations, we need to define a set of $K$ cluster centroids (e.g., prototypical representations) $\{c_k\}_{k=1}^K$, which reside in the same representation space $S$. We

then *jointly* optimize $\mathcal{L}$ w.r.t $\theta$ and $\{c_k\}$ such that the representations $z(x_i)$ become structured in a way undisclosing *underlying probabilistic structures* in datasets.

We aim for higher probabilities $p$ for points closer to these centroids. To achieve this, we first define scores $\alpha_k(x_i) = -\|z(x_i) - c_k\|^2$ to quantify the compatibility between representations and centroids $c_k$, and then transform them into cluster membership probabilities using softmax, yielding *soft* cluster assignments $p_k(x_i) = \frac{\exp(\alpha_k(x_i))}{\sum_{j=1}^{K} \exp(\alpha_j(x_i))}$. Note that the ground-truth cluster assignments $y_i$ are known during pre-training, and hence, the maximum likelihood centroids can be straightforwardly estimated as $\hat{c}_k = \frac{1}{N_k} \sum_{\{i:y_i=k\}} z(x_i)$, where $N_k$ denotes the number of data points assigned to the $k$-th cluster. Following this formulation, the optimization needs to be performed only w.r.t the parameters $\theta$ of the neural network $f_\theta$, as the centroids are already specified by them.

Besides the centroids, we additionally consider another concept from GMMs, namely cluster priors $\pi_k$. Similar to centroids, the prior cluster frequencies $\hat{\pi}_k$ can be estimated from the training labels as the proportions of examples in the $k$-th cluster, and then incorporated directly into the probability calculation, resulting in the final form of our probabilistic scoring:

$$p_k(x_i) = \frac{\hat{\pi}_k \exp(\alpha_k(x_i))}{\sum_{j=1}^{K} \hat{\pi}_j \exp(\alpha_j(x_i))}, \quad where \quad \alpha_k(x_i) = -\|z(x_i) - \hat{c}_k\|^2. \tag{3}$$

**Remark 2.1.** *ZEUS cluster assignments correspond to the membership probabilities in GMMs defined as:* $r_{ik} = \frac{\pi_k \mathcal{N}(z(x_i)|c_k, \Sigma_k)}{\sum_{j=1}^{K} \pi_j \mathcal{N}(z(x_i)|c_j, \Sigma_j)}$, *but with fixed diagonal covariance matrices* $\Sigma_k := I$, *effectively forcing the encoder to learn circular clusters rather than elliptical ones.*

**Inference.** The structural relations learned during pre-training are leveraged during inference when the representations $z$ produced by the model $f_{\theta^*}$ are used to create a predictive distribution for new, previously unseen inputs $\mathbf{x}^*$ as $\hat{\mathbf{p}}^* := p(\mathbf{y}^* \mid f_{\theta^*}(\mathbf{x}^*))$. At this point, the labels $\mathbf{y}^*$ are unknown and therefore cannot be used to estimate the centroids nor the priors. Hence, to structure the obtained representation into clusters, we use a traditional learning algorithm, for example, similar to GMMs relying on Expectation-Maximization or simply k-means.

**Regularization.** Although theoretically $\mathcal{L}_{prob}$ is sufficient to build a clustering representation, we experimentally verified that the introduction of additional regularizers further improves the predictions in the inference (see Section 3.4).

First, we aim to ensure that the representations $z(x_i)$ associated with a particular cluster are compactly distributed around their corresponding centroid to enhance intra-cluster cohesion. We define a point concentration regularizer that explicitly minimizes the distance between representations and the centroids, analogous to the k-means objective of minimizing within-cluster sum of squares:

$$\mathcal{L}_{cp} = \sum_{k} \sum_{i:y_i=k} \alpha_k(x_i). \tag{4}$$

Second, to prevent the cluster centroids $c_k$ from collapsing towards similar points in the space, we add a centroid separation regularizer. We achieve this by maximizing the sum of squared distances between all distinct pairs of centroids. However, to avoid this term dominating the loss if centroids were pushed infinitely far apart, we cap the contribution of each pair's squared distance at a predefined threshold $T$. The term to be minimized is thus:

$$\mathcal{L}_{sep} = -\sum_{k=1}^{K} \sum_{j=k+1}^{K} \min(\|\hat{c}_k - \hat{c}_j\|^2, T). \tag{5}$$

The *final loss* combines the main clustering objective with two regularization terms as

$$\mathcal{L} = \mathcal{L}_{prob} + \lambda_{cp}\mathcal{L}_{cp} + \lambda_{sep}\mathcal{L}_{sep} \tag{6}$$

where the hyperparameters $\lambda_{cp} \geq 0$ and $\lambda_{sep} \geq 0$ promote point concentration and control the relative importance of enforcing centroid separation, respectively. We used simply $\lambda_{cp} = \lambda_{sep} = 1$.

## 2.2 Prior data for pre-training ZEUS

The key component of ZEUS is the data-generating prior $p(\mathcal{D})$. As it primarily affects the generalization ability of the pre-trained model $f_{\theta^*}$, it must be designed to cover diverse data distributions and various cluster configurations. In particular, we construct it as a latent variable model (LVM), assuming that each dataset $\mathcal{D}$ is sampled from a $K$-component mixture of distributions. Formally, the probability of a data point $x \in \mathcal{D}$ under this model is

$$p(x) = \sum_{k=1}^{K} p(y = k)\, p(x \mid y = k),$$

where $p(y)$ is a categorical distribution, and $p(x \mid y = k)$ represents the $k$-th (continuous) component. Although the categories $y$ remain latent for real datasets, they are known during synthetic data generation and can serve as ground-truth labels – a property integral to pre-training ZEUS. This procedure, coupled with Theorem 1 (in Appendix) ensures that a sufficiently expressive model trained on synthetic datasets can handle real datasets drawn from arbitrary mixtures of distributions.

The number of categories $K$ we sample uniformly between 2 and 10, and the observations $x$ are sampled from multivariate Gaussian distributions. To control the complexity of datasets, we introduce a constraint (Eq. 7 in Appendix) that ensures a sufficient separation between each pair of components. For each component, we generate between 50 and 800 samples. Since real data rarely consist of Gaussian clusters, we additionally transform the data points from each category using randomized ResNet-like neural networks to produce more realistic cluster shapes. We selected ResNets due to their properties, as they can define invertible transformations [2], and this ensures that clustering structures will be preserved in outputs. Finally, we append a certain fraction of categorical features in one-hot encoding scheme to selected datasets, which is followed by an optional PCA reduction to keep the data dimension at the requested maximum level. Complete details of the prior-data generating process can be found in Appendix B.2. Theoretical justification of this procedure is presented in Appendix B.1.

The above strategy for generating clustering datasets illustrates the key idea behind ZEUS. Instead of clustering each dataset individually using an arbitrary loss criterion (as in $k$-means or DEC), ZEUS learns to perform clustering by inverting the data generation process. Although our synthetic datasets are limited by the selected family of distributions specified above, the proposed paradigm for zero-shot unsupervised learning for clustering is general.

## 2.3 Relation to Bayesian learning and Prior-Data Fitted Networks

The recently introduced framework of *Bayesian inference through transformers* demonstrates that neural networks pre-trained on synthetic datasets implicitly approximate Bayesian inference without explicitly computing posterior distributions [25, 16]. By framing the approach described in Section 2.1 as a Prior-Data Fitted Network (PFN), we show ZEUS implicitly performs approximate Bayesian averaging.

Given a prior distribution $p(\mathcal{D})$ over (synthetic) datasets $\mathcal{D}$, a PFN parameterized by $\theta$ is trained to minimize the negative log-likelihood of predicting held-out labels within datasets sampled from this prior. The associated loss function is defined as: $\mathcal{L}_{PFN}(\theta) = \mathbb{E}_{\mathcal{D}_{ctx} \cup \{(x,y)\} \sim p(\mathcal{D})}[-\log q_\theta(y|x, \mathcal{D}_{ctx})]$. Minimizing the Prior-Data Negative Log-Likelihood is then equivalent to minimizing the expected Kullback–Leibler divergence between the network's predictive distribution and the true Posterior Predictive Distribution (PPD) $p(y|x, \mathcal{D}_{ctx}) = \int_\Phi p(y|x, \phi)p(\mathcal{D}_{ctx}|\phi)p(\phi)d\phi$ (see Corollary 1.1 in [25]). A pre-trained PFN approximates this integral implicitly, yielding a distribution $q_\theta(y|x, \mathcal{D}_{ctx})$ directly from forward propagation of the network. In Section A, a more detailed explanation of PFNs was provided.

ZEUS instead of directly outputting $q_\theta(y|x, \mathcal{D}_{ctx})$, maps inputs $x$ to latent representation vectors $z(x)$. The probabilistic assignments $p_y(x)$ are then constructed from these vectors according to Eq. 3. This equation specifies a PPD for inference, but also defines a probability mass function for training:

**Remark 2.2.** *Eqs. 1, 2, and 3 constitute a valid Prior-Data Negative Log-Likelihood, equivalent to Eq. (2) from [25] with $\mathcal{D}_{ctx} = \emptyset$.*

This follows by mapping $p_y(x) := q_\theta(y|x, \emptyset)$ and noting that Eq. 2 corresponds to the cross-entropy between the true labels $\{y\}$ and the probabilistic assignments $p_y(x)$. Intuitively, this formulation

reinterprets probabilistic clustering with known labels as a classification task. Then, pre-training the transformer by minimizing the cross-entropy loss over prior-generated datasets remains identical to PFNs, and thus, we can conclude that ZEUS *implicitly learns a Bayesian approximation through prior fitting*.

Our method, however, deviates from traditional PFNs by imposing an explicit mixture-like structure on latent representations (Eq. 3), unlike more general PFNs:

**Remark 2.3.** *By enforcing the clustering structure, ZEUS may impose stronger assumptions on the PPD compared to vanilla PFNs. This structure might be suboptimal for classification tasks and the true PPD may not belong to the family of attainable solutions.*

As explained above, our clustering-based extension is theoretically sound, nonetheless, the assumption could limit representational expressivity compared to the more flexible transformers employed for the original PFNs [16]. On the other hand, the enforced structures shall be more appropriate for unsupervised tasks.

## 3 Experiments

This section presents an experimental evaluation of the clustering performance of our method. Due to space constraints, further results are provided in Appendix G.

### 3.1 Experimental setup

**Model architecture:** ZEUS relies on a transformer architecture similar to TabPFN [16]. It consists of 12 attention blocks, each with 6 heads and a token dimension of 512, with GeLU activation employed. Following the TabPFN design, each data point is first embedded using a linear transformation and then passed as a token to the transformer. For the unsupervised setting, no label embeddings are ever created or presented to the model. Similarly, for the zero-shot case, no query set is used either, and consequently, attention is computed solely over the support set. Finally, unlike TabPFN, we omit any additional MLP decoder after the transformer.

**Pre-training:** In the pre-training phase, we sample datasets from the mixture of Gaussians and transformed mixtures in equal proportions (1:1), generating 1000 unique dataset batch samples for each epoch. For training, we employ the Adam optimizer along with a cosine learning rate scheduler with warm-up, using a learning rate of 2e-5. The plot illustrating model improvements during the pre-training process is available in Section C.

**Inference:** During inference, preprocessing of each dataset involves standardizing numerical features, followed by scaling them to the range $[-1, 1]$, whereas categorical features are transformed using one-hot encoding. The input size of our model is fixed to 30. For datasets with lower dimensionality, we pad the missing positions with zeros, while for higher-dimensional datasets, we reduce the number of input features via Principal Component Analysis (PCA). Unless stated otherwise, at inference we use k-means applied to the normalized (=scaled to $[-1, 1]$) transformer output in order to obtain clusters from our learned representation.

**Datasets:** For evaluation, we consider three groups of datasets: real datasets from OpenML [3] (*Real*), synthetic mixtures of Gaussians (*Syn. Gauss.*), and synthetic mixtures of Gaussians transformed by ResNet-like neural networks (*Syn. Transf.*). Both types of synthetic datasets are augmented with categorical variables. The process of generating synthetic datasets is described in Section 2.2.

Each dataset contains at most 2000 samples as per model design and due to memory limitations. The study covers 34 real datasets, selected based on their clustering feasibility, defined as ARI $\geq 0.4$ achieved by at least one of the methods. Additionally, 20 synthetic datasets of each type were generated from the same prior as in the pre-training phase, but with a different random seed to ensure a fair comparison. Detailed statistics of the datasets, such as the number of numerical and categorical features, are provided in Appendix D.

**Baselines:** We compare ZEUS against a wide spectrum of state-of-the-art clustering methods used for tabular data. It includes $k$-means (KM), Gaussian Mixture Model (GMM), and deep-learning methods based on autoencoder (AE) architectures, including DEC [32], IDEC [13], IDC [30], and G-CEALS [27]. We additionally consider $k$-means and GMM performed in the autoencoder latent space

Table 1: Clustering quality (ARI) of ZEUS versus competing methods (higher is better).

| | KM | GMM | AE-KM | AE-GMM | DEC | IDEC | IDC | G-CEALS | TabPFN | SCARF | ZEUS |
|---|---|---|---|---|---|---|---|---|---|---|---|
| Real | 55.54 | 48.49 | 51.43 | 53.56 | 55.93 | 54.57 | 52.28 | 40.37 | 31.32 | 26.95 | **57.43** |
| Syn. Gauss. | **89.90** | 76.93 | 81.26 | 81.40 | 89.35 | 82.57 | 66.43 | 62.84 | 55.97 | 8.32 | 89.03 |
| Syn. Transf. | 75.04 | 75.88 | 60.45 | 71.29 | 79.94 | 61.26 | 66.78 | 49.17 | 15.66 | 2.48 | **86.33** |

Table 2: Average rank of the methods used in the benchmark (lower is better).

| | KM | GMM | AE-KM | AE-GMM | DEC | IDEC | IDC | G-CEALS | TabPFN | SCARF | ZEUS |
|---|---|---|---|---|---|---|---|---|---|---|---|
| Real | 4.69 | 5.65 | 5.72 | 5.24 | 4.69 | 5.01 | 5.62 | 8.18 | 8.22 | 8.85 | **4.13** |
| Syn. Gauss. | **2.65** | 3.65 | 5.65 | 5.60 | 3.23 | 5.70 | 7.95 | 8.90 | 8.75 | 11.00 | 2.92 |
| Syn. Transf. | 4.80 | 3.50 | 6.85 | 4.50 | 3.20 | 6.35 | 6.05 | 7.80 | 9.75 | 11.00 | **2.20** |

(respectively referred to as AE-KM and AE-GMM), along with $k$-means applied to representations obtained from TabPFN-Unsupervised [17] (TabPFN) and SCARF [1], to assess the clustering quality of different feature representations.

For AE-based baselines, we employ the standard configuration used in the prior literature, i.e., an architecture comprising of hidden layers with sizes [500, 500, 2000]. We used the latent dimension of 20, which we verified experimentally as the best value. Further details on the hyperparameters and code repositories for the baselines are provided in Appendix E.

**Reporting:** In the main text, we report only the results aggregated for each group of datasets (*Real*, *Syn. Gauss*, *Syn. Transf.*), while detailed results for individual datasets can be found in Section F. For the reader's convenience, we **bold** the best results and underline the second-best ones.

### 3.2 How effective is ZEUS for clustering?

To evaluate the performance of the clustering methods, we employ a standard evaluation procedure, where clusters identified by models are expected to correspond to undisclosed ground-truth classes. We use the Adjusted Rand Index (ARI) for quantitative evaluation, and to improve readability, we scale the ARI values by a factor of 100, where 100 indicates perfect clustering and values near 0 represent random grouping. Table 1 presents a summary of ARI scores averaged over 5 random seeds and all datasets within each dataset group. Additionally, Table 2 displays the average rankings for each of the methods.

We observe that ZEUS achieves competitive performance for all three groups of datasets. In particular, it achieves the best average ARI and the best rank for both the OpenML and Synthetic Transformed datasets, e.g., for the most challenging clustering scenarios. Notably, ZEUS outperforms the second-best method, DEC, by more than 6 percentage points, and the classical baselines by over 10 percentage points on the Synthetic Transformed datasets. The results on the Real datasets further demonstrate that ZEUS effectively generalizes the knowledge acquired during the pre-training stage on synthetic data to general data distributions. For the simplest Gaussian datasets, ZEUS ends up in top-3, with results only slightly below those of k-means and DEC. In terms of average rank it gets the second place.

Among the AE-based methods, DEC performs the best, while G-CEALS consistently demonstrates the weakest performance across

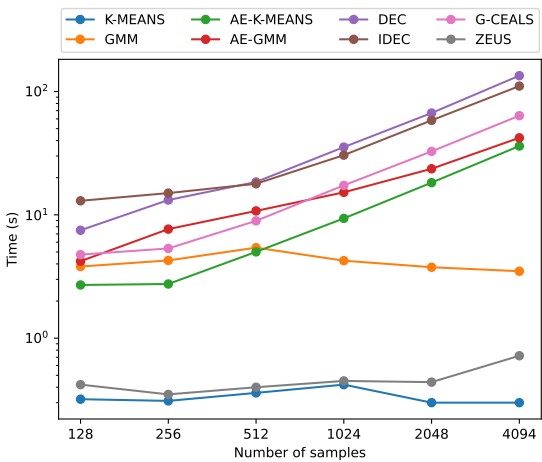

Figure 3: Clustering time vs. input size.

all datasets. K-means and GMM, as representatives of the classical methods, perform reasonably well.

Table 3: Soft clustering quality of ZEUS vs. baselines, measured by Brier score (lower is better).

| | KM | GMM | AE-KM | AE-GMM | DEC | IDEC | G-CEALS | ZEUS |
|---|---|---|---|---|---|---|---|---|
| Real | 0.4366 | 0.4799 | 0.4643 | 0.4679 | 0.3941 | **0.3671** | 0.4722 | 0.3817 |
| Syn. Gauss. | **0.0970** | 0.3110 | 0.2073 | 0.2484 | 0.3943 | 0.2308 | 0.3946 | 0.1269 |
| Syn. Transf. | 0.2803 | 0.2566 | 0.4535 | 0.3140 | 0.4638 | 0.3892 | 0.4951 | **0.1796** |

In particular, k-means consistently lands among the best-performing methods, which justifies its broad adoption among practitioners, despite being one of the most basic approaches. On the other hand, approaches such as SCARF or TabPFN are well-suited for tasks like classification, thanks to their strong performance in feature extraction. However, without additional regularization during training or specialized post-hoc fine-tuning (such as with DEC), effective separation of their representations is difficult. The detailed results, including scores for individual datasets corresponding to the averages in Tables 1 and 2, can be found in Appendix F.1

Finally, Figure 3 illustrates how the examined clustering methods scale with the increasing number of input data points. ZEUS maintains almost constant time while being only slightly slower than the basic k-means. It shows that the overhead from creating the representations by ZEUS is minimal. On the other hand, the remaining deep clustering algorithms require significantly more time and scale poorly with the increasing input size.

## 3.3 Are ZEUS's predictions calibrated?

For applications where uncertainty quantification is as crucial as the clustering decisions themselves, assessing calibration is particularly relevant. Having demonstrated the strong performance of ZEUS for hard clustering, we now examine whether its probabilistic foundations yield well-calibrated soft assignments. We compare it against competing approaches using the Brier score, which measures the accuracy of predicted probabilities. Unlike the previously used ARI, it penalizes both incorrect cluster assignments and *poorly calibrated confidence scores*, thereby providing a more comprehensive evaluation.

The Brier score is a supervised metric, meaning its direct computation for unsupervised clustering tasks is not straightforward. However, when ground-truth classes are available and the number of clusters matches the number of classes, a one-to-one mapping between clusters and classes can be established using the Hungarian algorithm [19, 20], which aims to maximize the total agreement between cluster-class pairs. The cost matrix $A$ for this assignment problem is $A_{jc} = \sum_{i=1}^{N} p_{ij} \cdot Y_{ic}$, where $Y_{ic} = 1$ if data point $i$ belongs to class $c$, and $Y_{ic} = 0$ otherwise.

Table 3 reports the average Brier score computed over 5 seeds for all datasets in respective groups. For the analysis, we used the vanilla variant of ZEUS, which is paired with the GMM clustering since k-means does not provide soft assignments. The covariances were constrained to be identities as implied by Eq. 3. All competing baselines, except for k-means, provide probabilistic cluster assignments, making the Brier score calculation straightforward for them. For k-means, we used one-hot encoding to represent its assignments as probabilities.

ZEUS demonstrates outstanding performance, achieving the best results for the Synthetic Transformed datasets while ranking second on both the OpenML and Synthetic Gaussian collections. The benefits of using ZEUS representations are especially visible in comparison with the vanilla GMM. Although the GMM achieves the second-best score on transformed data, its performance across the remaining datasets is merely modest. Among the remaining baselines, IDEC exhibits an interesting pattern: despite relatively weak clustering performance in Table 1, it achieves the top position on real datasets according to the Brier score and shows marked improvement on synthetic data. DEC displays the opposite tendency, with calibration results significantly inferior to its strong ARI performance. While G-CEALS remains generally weaker than other methods, its calibration performance consistently exceeds its clustering results presented in Table 1. The K-means algorithm, both in its standard implementation and when applied to autoencoder embeddings, yields impressive scores on Gaussian-categorical mixture datasets – likely due to the inherent separability of these data structures, which allows even simple one-hot probability estimates to produce well-calibrated predictions.

### 3.4 How helpful is regularization for ZEUS?

Representations obtained by minimizing the basic loss $\mathcal{L}_{prob}$ can be further improved by regularizing the optimization process to structure the latent representations. In Table 4, we examine how different combinations of the $\mathcal{L}_{prob}$ loss with the regularizers $\mathcal{L}_{cp}$ and $\mathcal{L}_{sep}$ affect the final performance across various datasets. To ensure a fair comparison, all models were evaluated using a data generator with fixed (identical for all) settings.

Table 4: Regularisation impact, measured by ARI (higher is better).

|  | $\mathcal{L}_{prob}$ | $\mathcal{L}_{prob} + \mathcal{L}_{sep}$ | $\mathcal{L}_{prob} + \mathcal{L}_{cp}$ | $\mathcal{L}_{prob} + \mathcal{L}_{sep} + \mathcal{L}_{cp}$ |
|---|---|---|---|---|
| Real | 44.80 | 51.60 | 48.65 | **57.43** |
| Syn. Gauss. | 83.37 | 81.88 | **90.59** | 89.03 |
| Syn. Transf. | 79.85 | 79.29 | **88.58** | 86.33 |

Our main model, which during pre-training incorporates both regularizers, exhibits a clear performance advantage on real datasets and consistently holds second place for the synthetic ones. Table 4 shows that a model pre-trained using only $\mathcal{L}_{prob}$ and its variant with $\mathcal{L}_{sep}$ included are insufficient for properly separating the transformer's representations for the clustering task, as they both struggle with clustering the synthetic datasets encountered during pre-training. On the other hand, the results for $\mathcal{L}_{prob} + \mathcal{L}_{cp}$ imply a positive impact by the compact loss component, $\mathcal{L}_{cp}$, especially for synthetic data, which, however, does not translate to strong performance on the OpenML benchmark. We conclude that all three loss components are necessary for good performance, with $\mathcal{L}_{cp}$ being crucial for pre-training and $\mathcal{L}_{sep}$ important for real data.

This mismatch between performance on real and synthetic datasets suggests a potential prior misspecification, which is then mitigated by the regularizers. In particular, among the real datasets, there may be some that are not well explained by the data generation process used during pre-training. Hence, for future work, one may want to explore alternative data-generating priors that may be more appropriate for these outliers.

### 3.5 What data-generating prior is optimal?

As mentioned in Section 2.2, constructing an appropriate data-generating prior $p(D)$ is crucial for maximizing performance and generalizability of ZEUS. To validate this claim, we conduct an ablation study evaluating four prior designs: *Gaussian + Categorical*, *NN-transformed + Categorical*, *Gaussian + NN-transformed*, and *Gaussian + NN-transformed + Categorical* (the standard ZEUS model). Each configuration name directly reflects the combination of priors used for pre-training (see Section B.2 in the Appendix for further details). For a fair comparison, we keep the default hyperparameter setup unchanged, except for the prior itself.

Table 5: ARI scores (averages; in rows) for ZEUS variants pre-trained on different priors (columns).

|  | Gauss. + Cat. | NN-transf. + Cat. | Gauss. + NN-transf. | Gauss. + NN-transf. + Cat. |
|---|---|---|---|---|
| Real | 40.59 | 50.90 | 52.00 | **57.43** |
| Syn. Gauss. | **92.61** | 89.90 | 75.25 | 89.03 |
| Syn. Transf. | 73.34 | **87.04** | 71.29 | 86.33 |

The results of the study are summarized in Table 5. As expected, the *Gaussian + Categorical* model achieves top performance on the Synthetic Gaussian datasets, while the *NN-transformed + Categorical* model excels for the Synthetic Transformed data. Notably, the latter also performs well on the Gaussian sets, as its prior is built upon transformed Gaussian mixtures. In contrast, the *Gaussian + NN-transformed* model yields relatively poor average ARI scores across both synthetic benchmarks. This drop is primarily caused by a significant decline in ARI performance on the categorical datasets. Nonetheless, its average rank remains consistently competitive. Finally, for the OpenML datasets, the best performance is achieved when pre-training includes all three types of priors, underscoring the complementary importance of each prior in the training process.

# 4 Related work

**Tabular data clustering.**  Traditional clustering algorithms, like k-means [23], GMM [7], or hierarchical methods [26], have widespread applications across data mining, bioinformatics, customer segmentation, and anomaly detection. However, these methods often rely on predefined distance metrics and fail to capture complex, non-linear relationships, making them suboptimal for high-dimensional and heterogeneous tabular datasets.

A pioneering Deep Embedded Clustering (DEC) [32] improves the target data representation by training the autoencoder and computing soft assignments in its latent space via Student's $t$-distribution. In a concurrent work [33], the authors perform joint dimensionality reduction using AE and k-means clustering in the latent space. Then, improved DEC (IDEC) [13] extends DEC by jointly optimizing reconstruction and clustering objectives, while spectral variants replace k-means steps with graph-based updates [8]. G-CEALS [27] replaces the $t$–distribution assumption with multivariate Gaussian clusters. DEPICT [10] attaches a softmax layer on top of an embedding network and trains with cross-entropy loss to eliminate the assumption of an explicit distribution prior. Finally, IDC [30] predicts interpretable cluster assignments at the instance and cluster levels.

Most of these deep learning models require careful hyperparameter tuning and early stopping which is unrealistic in the fully unsupervised setting due to the lack of labels [28, 29]. Moreover, the optimization process has to be performed for each dataset, which is often time-consuming. Although multiple deep clustering approaches are currently in use, most of them are designed for texts or images [37, 5, 22] and cannot be directly adapted for tabular data due to the lack of a dominant neural architecture for heterogeneous tabular inputs [12, 24].

**Representation learning for tabular data.**  Self-supervised learning (SSL) has been transformative in domains like vision and language [6, 11] but has struggled to show similar success in tabular data [34, 36]. A large diversity of data and a lack of pre-defined correlation between features hinder the design of universal pretext tasks or augmentations, as well as the transfer learning between domains [35]. On the other hand, in-context and zero-shot learning [4] enable the use of a single pre-trained model for multiple tasks out of the box, without any additional tuning. In particular, TabPFN treats small tabular datasets as contexts consisting of features along with labels, and achieves state-of-the-art classification in a single forward pass [16]. Although theoretical analyses reveal that transformers can implicitly implement algorithms such as gradient descent *in context*, their use is currently restricted to supervised problems [9].

# 5 Conclusion

In this paper, we presented ZEUS, a zero-shot transformer-based model that enables effective and efficient clustering of tabular data without the need for fine-tuning or extensive hyperparameter search. By pre-training on synthetic datasets with known latent structures, ZEUS learns generalizable representations that help simple clustering algorithms to uncover meaningful structures. Our experiments show that ZEUS consistently matches or outperforms both classical and deep learning-based clustering methods, offering a practical and theoretically grounded solution for unsupervised analysis of tabular data.

**Limitations.**  Since ZEUS is technically based on the TabPFN architecture, it inherits some of its drawbacks: a maximum number of input features and samples. However, the recently introduced TabPFN v2 [17] showed that tabular transformers can process larger datasets with a negligible increase in computational time. Moreover, our experimental results demonstrate that even if the dimension of input data exceeds the fixed value of 30, applying PCA does not significantly hurt the clustering performance.

The final performance of ZEUS heavily relies on the synthetic data used in pre-training. While we release a basic version of ZEUS, which encodes certain assumptions about data clusters, one can adjust the pre-training stage using different datasets. Finally, ZEUS does not cluster data itself, but constructs a convenient embedding space in which clustering can be performed using basic algorithms.

## Acknowledgments and Disclosure of Funding

We are grateful to the Reviewers for their effort and insightful comments on the paper.

This research is part of the project No. **2022/45/P/ST6/02969** co-funded by the National Science Centre and the European Union Framework Programme for Research and Innovation Horizon 2020 under the Marie Skłodowska-Curie grant agreement No. 945339. For the purpose of Open Access, the authors have applied a CC-BY public copyright licence to any Author Accepted Manuscript (AAM) version arising from this submission.

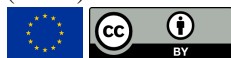

The research of P. Marszałek and M. Śmieja was supported by the National Science Centre (Poland), grant no. **2023/50/E/ST6/00169**. The research of Witold Wydmański was supported by the Ministry of Science grant no. **PN/01/0195/2022** and NCN Sonata BIS grant number **2020/38/E/NZ2/00598**. The research of J. Tabor was supported by the National Science Centre (Poland), grant no. **2023/49/B/ST6/01137**. Some experiments were performed on servers purchased with funds from the flagship project entitled "Artificial Intelligence Computing Center Core Facility" from the DigiWorld Priority Research Area within the Excellence Initiative – Research University program at Jagiellonian University in Krakow. We also gratefully acknowledge Polish high-performance computing infrastructure PLGrid (HPC Center: ACK Cyfronet AGH) for providing computer facilities and support within computational grant no. **PLG/2024/017893**.

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

# NeurIPS Paper Checklist

1. **Claims**

   Question: Do the main claims made in the abstract and introduction accurately reflect the paper's contributions and scope?

   Answer: [Yes]

   Justification: The abstract and introduction clearly state the main contributions: (1) the introduction of ZEUS, a transformer-based model for zero-shot clustering of tabular data; (2) the use of synthetic datasets with known latent-variable structure for pre-training; and (3) strong empirical results on both real and synthetic datasets. These claims are consistently substantiated in the methodology, experiments, and analysis sections. The scope and limitations (e.g., dimensionality constraints, reliance on synthetic priors) are also acknowledged in the conclusion.

   Guidelines:
   - The answer NA means that the abstract and introduction do not include the claims made in the paper.
   - The abstract and/or introduction should clearly state the claims made, including the contributions made in the paper and important assumptions and limitations. A No or NA answer to this question will not be perceived well by the reviewers.
   - The claims made should match theoretical and experimental results, and reflect how much the results can be expected to generalize to other settings.
   - It is fine to include aspirational goals as motivation as long as it is clear that these goals are not attained by the paper.

2. **Limitations**

   Question: Does the paper discuss the limitations of the work performed by the authors?

   Answer: [Yes]

   Justification: The paper includes a dedicated Limitations paragraph in the conclusion, where it explicitly discusses constraints related to model capacity (e.g., fixed number of features and samples inherited from TabPFN), reliance on synthetic data for pre-training, and the fact that ZEUS generates embeddings but does not perform clustering end-to-end. It also notes the potential for prior misspecification and suggests that the performance may vary depending on how well the synthetic priors match real-world data. These limitations are realistic, clearly acknowledged, and contextualized in terms of potential extensions.

   Guidelines:
   - The answer NA means that the paper has no limitation while the answer No means that the paper has limitations, but those are not discussed in the paper.
   - The authors are encouraged to create a separate "Limitations" section in their paper.
   - The paper should point out any strong assumptions and how robust the results are to violations of these assumptions (e.g., independence assumptions, noiseless settings, model well-specification, asymptotic approximations only holding locally). The authors should reflect on how these assumptions might be violated in practice and what the implications would be.
   - The authors should reflect on the scope of the claims made, e.g., if the approach was only tested on a few datasets or with a few runs. In general, empirical results often depend on implicit assumptions, which should be articulated.
   - The authors should reflect on the factors that influence the performance of the approach. For example, a facial recognition algorithm may perform poorly when image resolution is low or images are taken in low lighting. Or a speech-to-text system might not be used reliably to provide closed captions for online lectures because it fails to handle technical jargon.
   - The authors should discuss the computational efficiency of the proposed algorithms and how they scale with dataset size.
   - If applicable, the authors should discuss possible limitations of their approach to address problems of privacy and fairness.

- While the authors might fear that complete honesty about limitations might be used by reviewers as grounds for rejection, a worse outcome might be that reviewers discover limitations that aren't acknowledged in the paper. The authors should use their best judgment and recognize that individual actions in favor of transparency play an important role in developing norms that preserve the integrity of the community. Reviewers will be specifically instructed to not penalize honesty concerning limitations.

3. **Theory assumptions and proofs**

   Question: For each theoretical result, does the paper provide the full set of assumptions and a complete (and correct) proof?

   Answer: [NA]

   Justification: The paper does not present formal theorems, lemmas, or propositions with accompanying proofs. While it includes a theoretical framing (e.g., the connection to Prior-Data Fitted Networks and Bayesian inference), these are presented as conceptual motivations rather than formal theoretical contributions. There are no formal mathematical results stated or proven.

   Guidelines:

   - The answer NA means that the paper does not include theoretical results.
   - All the theorems, formulas, and proofs in the paper should be numbered and cross-referenced.
   - All assumptions should be clearly stated or referenced in the statement of any theorems.
   - The proofs can either appear in the main paper or the supplemental material, but if they appear in the supplemental material, the authors are encouraged to provide a short proof sketch to provide intuition.
   - Inversely, any informal proof provided in the core of the paper should be complemented by formal proofs provided in appendix or supplemental material.
   - Theorems and Lemmas that the proof relies upon should be properly referenced.

4. **Experimental result reproducibility**

   Question: Does the paper fully disclose all the information needed to reproduce the main experimental results of the paper to the extent that it affects the main claims and/or conclusions of the paper (regardless of whether the code and data are provided or not)?

   Answer: [Yes]

   Justification: The paper provides comprehensive details necessary for reproducing the main results. It specifies the model architecture, the training procedure (data generation process, optimizer, learning rate, batch size, number of epochs), the inference pipeline (standardization, PCA, normalization, clustering algorithm), and the datasets used (OpenML datasets, synthetic datasets with known priors). It also includes evaluation metrics, and explicit descriptions of ablation setups. We also provide a link to an anonymized code repository.

   Guidelines:

   - The answer NA means that the paper does not include experiments.
   - If the paper includes experiments, a No answer to this question will not be perceived well by the reviewers: Making the paper reproducible is important, regardless of whether the code and data are provided or not.
   - If the contribution is a dataset and/or model, the authors should describe the steps taken to make their results reproducible or verifiable.
   - Depending on the contribution, reproducibility can be accomplished in various ways. For example, if the contribution is a novel architecture, describing the architecture fully might suffice, or if the contribution is a specific model and empirical evaluation, it may be necessary to either make it possible for others to replicate the model with the same dataset, or provide access to the model. In general. releasing code and data is often one good way to accomplish this, but reproducibility can also be provided via detailed instructions for how to replicate the results, access to a hosted model (e.g., in the case of a large language model), releasing of a model checkpoint, or other means that are appropriate to the research performed.

- While NeurIPS does not require releasing code, the conference does require all submissions to provide some reasonable avenue for reproducibility, which may depend on the nature of the contribution. For example
  (a) If the contribution is primarily a new algorithm, the paper should make it clear how to reproduce that algorithm.
  (b) If the contribution is primarily a new model architecture, the paper should describe the architecture clearly and fully.
  (c) If the contribution is a new model (e.g., a large language model), then there should either be a way to access this model for reproducing the results or a way to reproduce the model (e.g., with an open-source dataset or instructions for how to construct the dataset).
  (d) We recognize that reproducibility may be tricky in some cases, in which case authors are welcome to describe the particular way they provide for reproducibility. In the case of closed-source models, it may be that access to the model is limited in some way (e.g., to registered users), but it should be possible for other researchers to have some path to reproducing or verifying the results.

5. **Open access to data and code**

   Question: Does the paper provide open access to the data and code, with sufficient instructions to faithfully reproduce the main experimental results, as described in supplemental material?

   Answer: [Yes]

   Justification: The paper includes a link to an anonymized code repository. The repository provides all necessary components to reproduce the main results: the model definition, synthetic dataset generator, training pipeline, preprocessing scripts, and evaluation code. Real-world datasets are sourced from OpenML (publicly accessible), and synthetic datasets can be generated using the provided scripts.

   Guidelines:
   - The answer NA means that paper does not include experiments requiring code.
   - Please see the NeurIPS code and data submission guidelines (`https://nips.cc/public/guides/CodeSubmissionPolicy`) for more details.
   - While we encourage the release of code and data, we understand that this might not be possible, so "No" is an acceptable answer. Papers cannot be rejected simply for not including code, unless this is central to the contribution (e.g., for a new open-source benchmark).
   - The instructions should contain the exact command and environment needed to run to reproduce the results. See the NeurIPS code and data submission guidelines (`https://nips.cc/public/guides/CodeSubmissionPolicy`) for more details.
   - The authors should provide instructions on data access and preparation, including how to access the raw data, preprocessed data, intermediate data, and generated data, etc.
   - The authors should provide scripts to reproduce all experimental results for the new proposed method and baselines. If only a subset of experiments are reproducible, they should state which ones are omitted from the script and why.
   - At submission time, to preserve anonymity, the authors should release anonymized versions (if applicable).
   - Providing as much information as possible in supplemental material (appended to the paper) is recommended, but including URLs to data and code is permitted.

6. **Experimental setting/details**

   Question: Does the paper specify all the training and test details (e.g., data splits, hyperparameters, how they were chosen, type of optimizer, etc.) necessary to understand the results?

   Answer: [Yes]

   Justification: The paper provides extensive details about the experimental setup in Section 3. It specifies:
   - Model architecture

- Optimizer and learning rate, as well as use of cosine scheduling with warm-up
- Dataset preprocessing steps (standardization, PCA for high-dimensional inputs, one-hot for categorical features)
- Evaluation metrics (ARI and Brier score)

These details are enough to interpret the results and, if needed, to reproduce them. Additional implementation details are provided in the appendix.

Guidelines:

- The answer NA means that the paper does not include experiments.
- The experimental setting should be presented in the core of the paper to a level of detail that is necessary to appreciate the results and make sense of them.
- The full details can be provided either with the code, in appendix, or as supplemental material.

7. **Experiment statistical significance**

Question: Does the paper report error bars suitably and correctly defined or other appropriate information about the statistical significance of the experiments?

Answer: [No]

Justification: While we mention that all results are averaged over 5 random seeds (e.g., in Tables 1, 3), we do not report any error bars, standard deviations, or confidence intervals. The source of variability (random seed) is implicitly mentioned, but no statistical significance tests or variability estimates are provided in figures or tables

Guidelines:

- The answer NA means that the paper does not include experiments.
- The authors should answer "Yes" if the results are accompanied by error bars, confidence intervals, or statistical significance tests, at least for the experiments that support the main claims of the paper.
- The factors of variability that the error bars are capturing should be clearly stated (for example, train/test split, initialization, random drawing of some parameter, or overall run with given experimental conditions).
- The method for calculating the error bars should be explained (closed form formula, call to a library function, bootstrap, etc.)
- The assumptions made should be given (e.g., Normally distributed errors).
- It should be clear whether the error bar is the standard deviation or the standard error of the mean.
- It is OK to report 1-sigma error bars, but one should state it. The authors should preferably report a 2-sigma error bar than state that they have a 96% CI, if the hypothesis of Normality of errors is not verified.
- For asymmetric distributions, the authors should be careful not to show in tables or figures symmetric error bars that would yield results that are out of range (e.g. negative error rates).
- If error bars are reported in tables or plots, The authors should explain in the text how they were calculated and reference the corresponding figures or tables in the text.

8. **Experiments compute resources**

Question: For each experiment, does the paper provide sufficient information on the computer resources (type of compute workers, memory, time of execution) needed to reproduce the experiments?

Answer: [No]

Justification: While the paper presents extensive experimental results, it does not specify the total compute cost.

Guidelines:

- The answer NA means that the paper does not include experiments.
- The paper should indicate the type of compute workers CPU or GPU, internal cluster, or cloud provider, including relevant memory and storage.

- The paper should provide the amount of compute required for each of the individual experimental runs as well as estimate the total compute.
- The paper should disclose whether the full research project required more compute than the experiments reported in the paper (e.g., preliminary or failed experiments that didn't make it into the paper).

9. **Code of ethics**

Question: Does the research conducted in the paper conform, in every respect, with the NeurIPS Code of Ethics https://neurips.cc/public/EthicsGuidelines?

Answer: [Yes]

Justification: The research focuses on unsupervised learning for tabular data using synthetic datasets and publicly available real-world datasets from OpenML. It does not involve any human subjects, sensitive personal data, or models with known societal or environmental risks. The synthetic data is generated in a controlled, fully artificial manner, and real datasets are used strictly for benchmarking in accordance with their licenses. The research is aligned with the NeurIPS Code of Ethics regarding reproducibility, transparency, and responsible development of machine learning models.

Guidelines:

- The answer NA means that the authors have not reviewed the NeurIPS Code of Ethics.
- If the authors answer No, they should explain the special circumstances that require a deviation from the Code of Ethics.
- The authors should make sure to preserve anonymity (e.g., if there is a special consideration due to laws or regulations in their jurisdiction).

10. **Broader impacts**

Question: Does the paper discuss both potential positive societal impacts and negative societal impacts of the work performed?

Answer: [Yes]

Justification: The paper includes a Broader Impact section in the appendix, which discusses both the potential benefits and risks of using ZEUS.

Guidelines:

- The answer NA means that there is no societal impact of the work performed.
- If the authors answer NA or No, they should explain why their work has no societal impact or why the paper does not address societal impact.
- Examples of negative societal impacts include potential malicious or unintended uses (e.g., disinformation, generating fake profiles, surveillance), fairness considerations (e.g., deployment of technologies that could make decisions that unfairly impact specific groups), privacy considerations, and security considerations.
- The conference expects that many papers will be foundational research and not tied to particular applications, let alone deployments. However, if there is a direct path to any negative applications, the authors should point it out. For example, it is legitimate to point out that an improvement in the quality of generative models could be used to generate deepfakes for disinformation. On the other hand, it is not needed to point out that a generic algorithm for optimizing neural networks could enable people to train models that generate Deepfakes faster.
- The authors should consider possible harms that could arise when the technology is being used as intended and functioning correctly, harms that could arise when the technology is being used as intended but gives incorrect results, and harms following from (intentional or unintentional) misuse of the technology.
- If there are negative societal impacts, the authors could also discuss possible mitigation strategies (e.g., gated release of models, providing defenses in addition to attacks, mechanisms for monitoring misuse, mechanisms to monitor how a system learns from feedback over time, improving the efficiency and accessibility of ML).

11. **Safeguards**

Question: Does the paper describe safeguards that have been put in place for responsible release of data or models that have a high risk for misuse (e.g., pre-trained language models, image generators, or scraped datasets)?

Answer: [NA]

Justification: ZEUS's pretraining utilizes exclusively synthetic datasets, and thus doesn't introduce any risk of misuse. It is not capable of generating human-like content, nor does it process natural language, images, or other modalities associated with misuse risk. As such, there is no high-risk component requiring safeguards.

Guidelines:

- The answer NA means that the paper poses no such risks.
- Released models that have a high risk for misuse or dual-use should be released with necessary safeguards to allow for controlled use of the model, for example by requiring that users adhere to usage guidelines or restrictions to access the model or implementing safety filters.
- Datasets that have been scraped from the Internet could pose safety risks. The authors should describe how they avoided releasing unsafe images.
- We recognize that providing effective safeguards is challenging, and many papers do not require this, but we encourage authors to take this into account and make a best faith effort.

12. **Licenses for existing assets**

Question: Are the creators or original owners of assets (e.g., code, data, models), used in the paper, properly credited and are the license and terms of use explicitly mentioned and properly respected?

Answer: [Yes]

Justification: The paper cites and builds upon the publicly available TabPFN v1 codebase, which is correctly credited and stated to be released under the Apache 2.0 License. All real-world datasets are obtained from OpenML.org and are explicitly described as open data, with the acknowledgment that individual datasets may have licenses such as CC-BY. This information is clearly documented in the paper in the "Licensing and Third-Party Assets" section of the appendix. The code released by the authors is also provided under the Apache 2.0 License, and appropriate instructions for reproduction are included.

Guidelines:

- The answer NA means that the paper does not use existing assets.
- The authors should cite the original paper that produced the code package or dataset.
- The authors should state which version of the asset is used and, if possible, include a URL.
- The name of the license (e.g., CC-BY 4.0) should be included for each asset.
- For scraped data from a particular source (e.g., website), the copyright and terms of service of that source should be provided.
- If assets are released, the license, copyright information, and terms of use in the package should be provided. For popular datasets, `paperswithcode.com/datasets` has curated licenses for some datasets. Their licensing guide can help determine the license of a dataset.
- For existing datasets that are re-packaged, both the original license and the license of the derived asset (if it has changed) should be provided.
- If this information is not available online, the authors are encouraged to reach out to the asset's creators.

13. **New assets**

Question: Are new assets introduced in the paper well documented and is the documentation provided alongside the assets?

Answer: [Yes]

Justification: The paper introduces a new model, ZEUS, and a synthetic data generation pipeline used for its pre-training. Both are made available in the anonymized code repository linked in the main text. The repository includes:

- code for the model architecture, training loop, and inference,
- scripts for generating synthetic datasets,
- configuration files for hyperparameters and preprocessing,
- usage instructions (e.g., via a README file).

Guidelines:

- The answer NA means that the paper does not release new assets.
- Researchers should communicate the details of the dataset/code/model as part of their submissions via structured templates. This includes details about training, license, limitations, etc.
- The paper should discuss whether and how consent was obtained from people whose asset is used.
- At submission time, remember to anonymize your assets (if applicable). You can either create an anonymized URL or include an anonymized zip file.

14. **Crowdsourcing and research with human subjects**

Question: For crowdsourcing experiments and research with human subjects, does the paper include the full text of instructions given to participants and screenshots, if applicable, as well as details about compensation (if any)?

Answer: [NA]

Justification: The paper does not involve any crowdsourcing or research with human subjects. All experiments are conducted on synthetic datasets or publicly available tabular datasets from OpenML, with no human annotation, survey, or data collection involved.

Guidelines:

- The answer NA means that the paper does not involve crowdsourcing nor research with human subjects.
- Including this information in the supplemental material is fine, but if the main contribution of the paper involves human subjects, then as much detail as possible should be included in the main paper.
- According to the NeurIPS Code of Ethics, workers involved in data collection, curation, or other labor should be paid at least the minimum wage in the country of the data collector.

15. **Institutional review board (IRB) approvals or equivalent for research with human subjects**

Question: Does the paper describe potential risks incurred by study participants, whether such risks were disclosed to the subjects, and whether Institutional Review Board (IRB) approvals (or an equivalent approval/review based on the requirements of your country or institution) were obtained?

Answer: [NA]

Justification: The paper does not involve neither crowdsourcing nor research with human subjects.

Guidelines:

- The answer NA means that the paper does not involve crowdsourcing nor research with human subjects.
- Depending on the country in which research is conducted, IRB approval (or equivalent) may be required for any human subjects research. If you obtained IRB approval, you should clearly state this in the paper.
- We recognize that the procedures for this may vary significantly between institutions and locations, and we expect authors to adhere to the NeurIPS Code of Ethics and the guidelines for their institution.

- For initial submissions, do not include any information that would break anonymity (if applicable), such as the institution conducting the review.

16. **Declaration of LLM usage**

    Question: Does the paper describe the usage of LLMs if it is an important, original, or non-standard component of the core methods in this research? Note that if the LLM is used only for writing, editing, or formatting purposes and does not impact the core methodology, scientific rigorousness, or originality of the research, declaration is not required.

    Answer: [NA]

    Justification: The research presented in this paper does not involve large language models (LLMs) as part of the methodology, experiments, or any original contribution. ZEUS is a transformer-based model trained on synthetic tabular data for zero-shot clustering, and all components are specific to structured data. No LLMs were used in model design, training, inference, evaluation, or theoretical framing. Therefore, no declaration under the NeurIPS LLM policy is required.

    Guidelines:

    - The answer NA means that the core method development in this research does not involve LLMs as any important, original, or non-standard components.
    - Please refer to our LLM policy (`https://neurips.cc/Conferences/2025/LLM`) for what should or should not be described.

# ZEUS: Zero-shot Embeddings for Unsupervised Separation of Tabular Data – supplementary material

## A  Background on Prior-Data Fitted Networks

The recently introduced framework of *Bayesian inference through transformers* demonstrates that neural networks pre-trained on synthetic datasets implicitly approximate Bayesian inference without explicitly computing posterior distributions [25, 16]. The pre-trained transformers are known as *Prior-Data Fitted Networks* (PFNs). The pre-training involves synthetic prior fitting, wherein a transformer network is trained offline on numerous datasets generated from a predefined prior distribution over tasks. This pre-training procedure allows the transformer to implicitly encode a Bayesian posterior predictive distribution by optimizing a cross-entropy loss between its predictions and the synthetic data labels.

Formally, pre-training is carried out as follows: given a prior distribution $p(\mathcal{D})$ over (synthetic) datasets $\mathcal{D}$, a PFN network parameterized by $\theta$ is trained to minimize the negative log-likelihood of predicting held-out labels within datasets sampled from this prior. The associated loss function, known as the Prior-Data Negative Log-Likelihood (Prior-Data NLL), is explicitly defined as:

$$\mathcal{L}_{PFN}(\theta) = \mathbb{E}_{\mathcal{D}_{ctx} \cup \{(x,y)\} \sim p(\mathcal{D})}[-\log q_\theta(y|x, \mathcal{D}_{ctx})],$$

where each dataset $D_{ctx}$ and data point $(x, y)$ are sampled from the predefined prior distribution $p(D)$. As shown formally in [25], minimizing this loss is mathematically equivalent to minimizing the expected cross-entropy between the predictive distribution $q_\theta(y|x, D_{ctx})$ and the true posterior predictive distribution (PPD) derived from the prior. Specifically, minimizing the Prior-Data NLL is equivalent to minimizing the expected Kullback–Leibler divergence between the network's predictive distribution and the true PPD $p(y|x, D_{ctx})$:

$$\mathcal{L}_{PFN}(\theta) = \mathbb{E}_{\mathcal{D}_{ctx}, x \sim p(D)}[H(p(\cdot|x, \mathcal{D}), q_\theta(\cdot|x, D_{ctx}))],$$

where $H$ denotes the cross-entropy. Thus, optimality of the predictive distribution $q_\theta$ implies matching it exactly to the true Bayesian posterior predictive distribution, provided that the parametric family of distributions defined by the transformer is sufficiently expressive [25].

Within this in-context learning as Bayesian inference framework, transformers approximate Bayesian averaging implicitly. Given a training dataset $\mathcal{D}_{ctx} = \{(x_i, y_i)\}_{i=1}^n$, a new input $x$, and a prior over hypotheses $\phi$, the Bayesian posterior predictive distribution is formally expressed as:

$$p(y|x, \mathcal{D}_{ctx}) = \int_\Phi p(y|x, \phi)p(\mathcal{D}_{ctx}|\phi)p(\phi)d\phi,$$

integrating over all hypotheses $\phi \in \Phi$. PFNs approximate this integral implicitly through pre-training on prior-generated datasets, yielding a distribution $q_\theta(y|x, \mathcal{D}_{ctx})$ directly from forward propagation of the network conditioned on dataset $\mathcal{D}_{ctx}$ [25, 16].

## B  Details of synthetic data generation process

### B.1  Theory for synthetic-to-real data generalization

**Theorem 1** (Univeral Approximation Theorem for Mixture Distributions). *Let $P = \sum_{i=1}^k \pi_i P_i$ be a mixture of distributions on $\mathbb{R}^d$, where each $P_i$ is a probability distribution, and $\pi_i > 0$, $\sum_{i=1}^k \pi_i = 1$.*

*Then there exists a mixture of Gaussians $Q = \sum_{i=1}^k \pi_i \mathcal{N}(\mu_i, \Sigma_i)$ and a neural network $F$ such that we can approximate $P$ with arbitrarily small error by the pushforward measure $F_\# Q$. Additionally, we can select $F$ so that $F_\# N(\mu_i, \Sigma_i)$ approximates $P_i$ with arbitrarily small error.*

*Proof.* We proceed in several steps.

**Step 1: Approximation of individual components.**
Making use of the universal approximation theorem, for each component $P_i$ there exists a neural network $F_i$ which transforms an arbitrary Gaussian $Q_i = \mathcal{N}(\mu_i, \Sigma_i)$ into $P_i$, i.e. $F_{i\#}Q_i \approx P_i$.

**Step 2: Constructing a unified neural network.**
We now define a single neural network $F$ that can represent all the $F_i$ networks. Define $F$ : $\mathbb{R}^d \times \{1, \ldots, k\} \to \mathbb{R}^d$ such that $F(z, i) \approx F_i(z)$. This can be implemented by encoding the index $i$ as a one-hot vector or embedding and concatenating it to $z$, enabling $F$ to learn the behavior of each $F_i$ conditioned on the index $i$. The network $F$ can be extended to the domain $\mathbb{R}^d \times \mathbb{R}$ in an arbitrary way.

**Step 3: Approximation of the full mixture.**
Since each $F_\#\mathcal{N}(\mu_i, \Sigma_i)$ approximates $P_i$, the mixture $F_\#Q = \sum_{i=1}^{k} \pi_i F_\#\mathcal{N}(\mu_i, \Sigma_i)$ approximates $P = \sum_{i=1}^{k} \pi_i P_i$.

Thus, we have constructed a mixture of Gaussians $Q$ and a neural network $F$ such that $F_\#Q$ approximates $P$, and each component $F_\#\mathcal{N}(\mu_i, \Sigma_i)$ approximates $P_i$ arbitrarily well. This completes the proof. $\qquad\square$

## B.2 Insights into data-generating priors

We use three types of probabilistic procedures to generate the prior data:

1. **Gaussian**. The goal is to create a continuous data type where each cluster follows a multivariate Gaussian distribution with carefully designed means and covariance matrices. This construction process is incremental. Starting with a mixture containing $k$ components, the addition of a new cluster involves initially placing it at position 0 and then shifting it in a randomly chosen direction. The covariance structure for each Gaussian is generated through eigendecomposition, with eigenvalues sampled from a predefined range of $[0.005, 0.05]$ to control the shape and orientation of the clusters. Additionally, to guarantee the presence of both full-rank Gaussians in higher dimensions and degenerate ones in lower dimensions, extra conditions are introduced to narrow the range of eigenvalues. These constraints are activated with probabilities of 0.25 and 0.2, respectively. To ensure adequate separation between clusters and to prevent trivial overlap, we apply a Wasserstein-2 distance constraint, requiring that the means $\mu_i$ and $\mu_j$ of any two clusters, along with their corresponding covariance matrices $\Sigma_i$ and $\Sigma_j$, are separated by at least a threshold $T$, as follows:

$$\|\mu_i - \mu_j\|_2^2 + tr(\Sigma_i + \Sigma_j - 2(\Sigma_i^{\frac{1}{2}} \Sigma_j \Sigma_i^{\frac{1}{2}})^{\frac{1}{2}}) \geq T \tag{7}$$

   The exact minimum distance value $T$ varies between 0.5 and 1.0 and is randomly selected for each component independently in order to promote data diversity. This entire approach allows us to model a variety of cluster geometries, from spherical to highly elongated elliptical shapes.

2. **Categorical**. To enrich the generated data, we incorporate categorical features alongside continuous ones by sampling from categorical distributions that are biased toward certain categories for each cluster. To produce these varied categorical probability patterns, we use the Dirichlet distribution. The probability of including categorical features is controlled by a $categorical\_chance$ parameter, set to 0.3. Up to 3 categorical variables may be added, each having between 2 and $max\_categories$ possible values, defined as 5. The resulting categorical variables are then converted to one-hot encoding and combined with the continuous features, producing mixed-type datasets that more closely resemble real-world tabular data.

3. **NN transformed**. To create more complex, non-linearly separable cluster structures, we apply transformations to the numerical features using random neural networks with 3 to 6 layers. These transformations map the original data through several non-linear operations while preserving cluster identity information, producing datasets with more challenging decision boundaries. Following the approach of Invertible Residual Networks [2], we constrain the spectral norm of each transformation layer to be less than 1. In addition, standardization is applied between residual layers to ensure more stable transformations. To help preserve cluster separability, we append one-hot vectors indicating the component identity to the numerical variables before passing them through the random neural network. After the transformation, these extra dimensions are removed using the PCA algorithm, restoring the data to its original dimensionality. This comprehensive approach enables the

simulation of intricate, non-linear data manifold structures often presented in real-world clustering problems without introducing degenerate configurations.

All features undergo standardization and scaling to ensure numerical stability during training. Continuous features are normalized to the range $[-1, 1]$, while ensuring that the relative separation between clusters is preserved.

By training on this diverse collection of synthetic datasets-ranging from well-separated Gaussian clusters to complex, transformed manifolds with mixed feature types ZEUS learns to identify meaningful cluster structures across a wide spectrum of data distributions. This enables it to adapt to previously unseen datasets at inference time without additional training.

## C    Pre-training plots

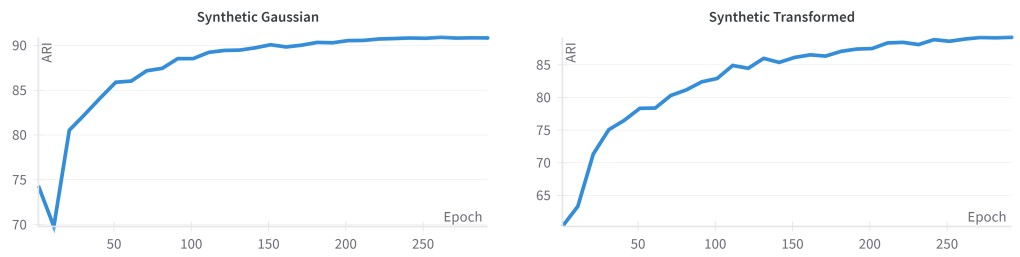

Figure 4: Visualization of pre-training process

Figure 4 presents two plots of average ARI over 200 synthetic validation datasets throughout 300 pre-training epochs. The first plot corresponds to the Gaussian-categorical datasets referred to as Synthetic Gaussian, while the second illustrates their NN transformed variants, named Synthetic Transformed. The data generation procedures are described in more detail in Appendix B.2. Both plots illustrate that the quality of the ZEUS representation improves over time, indicating that the model is learning new patterns, as evidenced by the increasing ARI during the pre-training process.

## D    Statistics of datasets used in experimental study

Tables 6, 7, and 8 provide detailed information about the datasets used in the experimental analysis. Each table includes the number of set instances, the number of categorical and numerical features, the total dimensionality, and the overall number of categories, represented by the length of the one-hot encoded vectors. Additionally, the tables report the number of classes that each dataset contains. In Table 6, the ID column corresponds to the OpenML ID, while in the remaining tables it functions solely as an index.

Table 6: Real/OpenML datasets statistics.

| ID | # Instances | # Numerical features | # Categorical features | Dimension | # Categories (one-hots) | # Classes |
|---|---|---|---|---|---|---|
| 14 | 2000 | 76 | 0 | 76 | 0 | 10 |
| 15 | 699 | 9 | 0 | 9 | 0 | 2 |
| 16 | 2000 | 64 | 0 | 64 | 0 | 10 |
| 18 | 2000 | 6 | 0 | 6 | 0 | 10 |
| 22 | 2000 | 47 | 0 | 47 | 0 | 10 |
| 35 | 366 | 1 | 33 | 130 | 129 | 6 |
| 51 | 294 | 6 | 7 | 25 | 19 | 2 |
| 53 | 270 | 13 | 0 | 13 | 0 | 2 |
| 56 | 435 | 0 | 16 | 32 | 32 | 2 |
| 61 | 150 | 4 | 0 | 4 | 0 | 3 |
| 187 | 178 | 13 | 0 | 13 | 0 | 3 |
| 377 | 600 | 60 | 0 | 60 | 0 | 6 |
| 458 | 841 | 70 | 0 | 70 | 0 | 4 |
| 481 | 209 | 7 | 1 | 14 | 7 | 2 |
| 694 | 310 | 8 | 0 | 8 | 0 | 9 |
| 721 | 200 | 10 | 0 | 10 | 0 | 2 |
| 733 | 209 | 6 | 0 | 6 | 0 | 2 |
| 745 | 159 | 14 | 1 | 20 | 6 | 2 |
| 756 | 159 | 15 | 0 | 15 | 0 | 2 |
| 796 | 209 | 6 | 1 | 36 | 30 | 2 |
| 820 | 235 | 12 | 0 | 12 | 0 | 2 |
| 840 | 205 | 17 | 8 | 68 | 51 | 2 |
| 854 | 158 | 5 | 2 | 14 | 9 | 2 |
| 1462 | 1372 | 4 | 0 | 4 | 0 | 2 |
| 1495 | 250 | 0 | 6 | 18 | 18 | 2 |
| 1499 | 210 | 7 | 0 | 7 | 0 | 3 |
| 1510 | 569 | 30 | 0 | 30 | 0 | 2 |
| 1523 | 310 | 6 | 0 | 6 | 0 | 3 |
| 4153 | 180 | 66 | 0 | 66 | 0 | 6 |
| 40496 | 500 | 7 | 0 | 7 | 0 | 10 |
| 40682 | 215 | 5 | 0 | 5 | 0 | 3 |
| 40705 | 959 | 42 | 2 | 44 | 2 | 2 |
| 42261 | 150 | 4 | 0 | 4 | 0 | 3 |
| 42585 | 344 | 4 | 2 | 10 | 6 | 3 |

Table 7: Synthetic Gaussian datasets statistics.

| ID | # Instances | # Numerical features | # Categorical features | Dimension | # Categories (one-hots) | # Classes |
|---|---|---|---|---|---|---|
| 0 | 1337 | 16 | 0 | 16 | 0 | 8 |
| 1 | 1383 | 23 | 0 | 23 | 0 | 8 |
| 2 | 1421 | 8 | 2 | 14 | 6 | 7 |
| 3 | 992 | 9 | 1 | 12 | 3 | 9 |
| 4 | 1314 | 8 | 0 | 8 | 0 | 9 |
| 5 | 1497 | 16 | 2 | 23 | 7 | 8 |
| 6 | 1370 | 2 | 0 | 2 | 0 | 6 |
| 7 | 1646 | 4 | 3 | 16 | 12 | 8 |
| 8 | 1520 | 11 | 0 | 11 | 0 | 8 |
| 9 | 1537 | 18 | 0 | 18 | 0 | 5 |
| 10 | 825 | 26 | 0 | 26 | 0 | 2 |
| 11 | 1112 | 9 | 0 | 9 | 0 | 5 |
| 12 | 1093 | 15 | 0 | 15 | 0 | 8 |
| 13 | 742 | 6 | 0 | 6 | 0 | 3 |
| 14 | 1595 | 11 | 2 | 18 | 7 | 7 |
| 15 | 1417 | 14 | 0 | 14 | 0 | 6 |
| 16 | 1787 | 28 | 0 | 28 | 0 | 5 |
| 17 | 764 | 19 | 0 | 19 | 0 | 4 |
| 18 | 889 | 25 | 0 | 25 | 0 | 5 |
| 19 | 1660 | 28 | 0 | 28 | 0 | 9 |

Table 8: Synthetic Transformed datasets statistics.

| ID | # Instances | # Numerical features | # Categorical features | Dimension | # Categories (one-hots) | # Classes |
|---|---|---|---|---|---|---|
| 0 | 1337 | 16 | 0 | 16 | 0 | 8 |
| 1 | 1627 | 30 | 0 | 30 | 0 | 9 |
| 2 | 1631 | 11 | 0 | 11 | 0 | 7 |
| 3 | 1891 | 15 | 1 | 17 | 2 | 10 |
| 4 | 1142 | 3 | 0 | 3 | 0 | 4 |
| 5 | 1222 | 24 | 0 | 24 | 0 | 9 |
| 6 | 953 | 6 | 0 | 6 | 0 | 6 |
| 7 | 1508 | 9 | 0 | 9 | 0 | 10 |
| 8 | 840 | 7 | 3 | 15 | 8 | 5 |
| 9 | 1745 | 14 | 0 | 14 | 0 | 9 |
| 10 | 1618 | 23 | 0 | 23 | 0 | 6 |
| 11 | 1432 | 13 | 0 | 13 | 0 | 9 |
| 12 | 1860 | 9 | 0 | 9 | 0 | 9 |
| 13 | 563 | 10 | 0 | 10 | 0 | 2 |
| 14 | 1033 | 6 | 2 | 13 | 7 | 3 |
| 15 | 750 | 2 | 0 | 2 | 0 | 4 |
| 16 | 1451 | 14 | 2 | 20 | 6 | 10 |
| 17 | 679 | 30 | 0 | 30 | 0 | 2 |
| 18 | 859 | 22 | 1 | 25 | 3 | 2 |
| 19 | 1493 | 11 | 0 | 11 | 0 | 7 |

# E   Baselines

The evaluation of baseline models is based on the following libraries and GitHub repositories:

1. **scikit-learn** - used for k-means and GMM,
2. **https://github.com/vlukiyanov/pt-dec** - implementation of the DEC,
3. **https://github.com/dawnranger/IDEC-pytorch** - source code for the IDEC,
4. **https://github.com/jsvir/idc** - official implementation of the IDC method,
5. **https://github.com/mdsamad001/G-CEALS—Deep-Clustering-for-Tabular-Data** - codebase for the GCEALS,
6. **https://github.com/PriorLabs/tabpfn-extensions** - a library that extends TabPFN functionality to a wide spectrum of machine learning tasks, including unsupervised ones,
7. **https://github.com/clabrugere/pytorch-scarf** - code repository for SCARF.

To ensure fair comparison, hyperparameters are chosen to maximize the performance of each method with respect to their overall average rank across 5 random seeds. For this reason, all numerical features are preprocessed using a standard scaler prior to the training phase.

Most parameters of the k-means and GMM methods are left at their default values. Only the $n\_init$ option was increased to 100 for k-means and 50 for GMM in order to improve stability.

As mentioned in Section 3.1, a standard autoencoder with hidden layers $[500, 500, 2000]$ is used for each AE-based method. The resulting network is first pre-trained for 1000 epochs and then fine-tuned for up to 1000 additional epochs, separately for each dataset and model. Following the GCEALS evaluation procedure, multiple latent dimension sizes $[5, 10, 15, 20]$ were tested across all considered methods. The results indicate that the default latent dimension of 10 does not yield the best performance; instead, a dimension of 20 generally performs better. Changing other hyperparameters, including the learning rate, optimizer, and clustering loss weight $\gamma$, generally did not lead to improved results. Therefore, the remaining parameters were left at their default values as proposed by the authors of the respective codebases.

For TabPFN, we use the *get_embeddings_per_column* method from the *TabPFNUnsupervisedModel* class, provided in the TabPFN Extensions repository[2], to extract per-column representations. These embeddings are then averaged across columns to obtain the final representation for each data point. For SCARF, we follow the procedure outlined in the *example.ipynb* notebook from the original repository[3]. In both cases, we cluster the learned representations using $k$-means, consistent with our approach in ZEUS.

# F   Extended experimental results

This section presents extended tables corresponding to the experiments described in Section 3.

## F.1   How effective is ZEUS for clustering?

Tables 9, 10, and 11 contain the complete results for individual datasets from the experiments discussed in Section 3.2 in Tables 1 and 2, as presented in the main text. The reported values represent averages of ARI over 5 random seeds. In addition to the rows presenting outcomes for specific datasets, the tables include summary rows labeled Mean, Mean-Rank, Top-3, and Top-1. These represent, respectively: the average ARI, the average rank computed across all datasets, the number of times a given model appears in the top 3, and the total number of wins achieved by each method. It is worth noting that the Top-1 and Top-3 rows indicate clear wins and clear appearances in top-3 positions.

The overall conclusions align with those presented in the primary analysis. One noteworthy point is that ZEUS achieves strong performance in terms of the Top-3 and Top-1 statistics, being outperformed in this regard only in Table 10, where k-means and GMM score better. Another interesting observation

---

[2]https://github.com/PriorLabs/tabpfn-extensions
[3]https://github.com/clabrugere/pytorch-scarf

is that GMM struggles with categorical features, which significantly worsens its average scores in Tables 10 and 11.

Table 9: Comparison of clustering quality using ARI metric on OpenML datasets (higher score indicate better performance).

| ID | KM | GMM | AE-KM | AE-GMM | DEC | IDEC | IDC | G-CEALS | TabPFN | SCARF | ZEUS |
|---|---|---|---|---|---|---|---|---|---|---|---|
| 14 | 38.74 | 45.90 | 40.94 | 45.36 | 49.60 | 36.78 | 45.03 | 45.04 | 22.02 | 1.90 | **50.56** |
| 15 | 82.85 | 71.38 | 61.44 | 71.74 | 86.29 | **87.48** | 83.20 | 24.38 | 74.31 | 75.01 | 81.28 |
| 16 | 55.62 | 64.75 | 61.66 | 69.97 | 68.56 | 55.61 | 65.84 | 55.27 | 14.69 | 3.74 | **74.03** |
| 18 | **54.86** | 46.88 | 51.45 | 51.47 | 50.00 | 53.54 | 44.40 | 49.93 | 51.50 | 5.02 | 51.63 |
| 22 | 35.48 | 47.39 | 31.09 | 50.80 | 50.02 | 35.69 | 46.03 | 28.38 | 11.09 | 2.63 | **56.05** |
| 35 | 70.68 | 70.68 | 70.22 | 70.48 | 68.63 | 64.58 | 77.99 | 56.57 | 52.24 | 74.61 | **85.12** |
| 51 | 28.35 | -2.54 | 37.82 | 34.13 | 38.13 | 34.58 | 38.93 | 35.13 | 38.26 | -1.27 | **43.66** |
| 53 | **45.21** | 5.17 | 39.14 | 40.62 | 43.50 | 41.91 | 33.03 | 21.32 | 26.61 | 24.24 | 35.76 |
| 56 | 57.79 | 58.49 | 57.10 | 58.31 | 56.97 | 55.20 | 59.25 | 55.66 | 58.42 | 61.26 | **66.41** |
| 61 | 62.01 | **90.39** | 59.79 | 60.21 | 57.84 | 60.14 | 58.30 | 51.37 | 77.20 | 51.19 | 85.15 |
| 187 | 89.75 | **93.09** | 82.87 | 84.86 | 85.02 | 85.10 | 80.04 | 55.67 | 37.55 | 40.05 | 88.19 |
| 377 | 56.70 | 58.47 | 62.06 | 59.52 | 63.03 | 59.01 | 62.39 | 59.79 | **66.89** | 14.45 | 55.30 |
| 458 | 95.08 | 96.97 | 64.33 | 75.70 | 94.58 | 63.29 | 86.77 | 68.31 | 61.41 | 2.87 | **99.19** |
| 481 | **58.51** | 2.77 | 36.84 | 48.03 | 57.28 | 52.93 | 43.17 | 19.67 | 2.77 | 1.79 | 8.68 |
| 694 | 35.91 | **43.30** | 35.26 | 36.56 | 26.30 | 36.02 | 33.47 | 30.76 | 27.68 | 14.09 | 33.38 |
| 721 | **43.28** | 43.28 | 13.12 | 25.12 | 17.75 | 28.77 | 12.74 | -0.03 | 23.63 | 6.35 | 43.28 |
| 733 | 50.97 | 46.29 | 38.39 | 55.11 | 67.47 | 73.51 | 68.88 | 26.26 | -8.35 | 49.41 | **74.97** |
| 745 | 57.63 | 55.44 | **78.35** | 75.52 | 46.37 | 50.94 | 59.29 | 57.81 | 1.26 | 16.75 | 73.85 |
| 756 | 55.70 | 4.46 | **72.81** | 53.40 | 58.60 | 53.32 | 44.14 | 10.60 | 64.82 | 50.31 |
| 796 | 49.62 | 2.26 | 76.54 | 52.19 | 50.07 | 55.92 | **79.59** | 43.40 | 0.88 | 58.25 | 13.99 |
| 820 | 50.16 | 36.77 | 44.29 | 49.24 | **51.21** | 49.30 | 44.59 | 24.51 | 50.26 | 13.04 | 28.87 |
| 840 | 39.05 | 48.37 | **53.60** | 46.72 | 49.14 | 39.16 | 19.59 | 27.95 | 8.89 | 32.13 | 27.66 |
| 854 | **76.14** | -0.21 | 62.33 | 60.83 | 35.05 | 70.52 | 64.81 | 72.25 | -0.54 | 6.48 | 76.14 |
| 1462 | 1.32 | 0.31 | 10.03 | 10.17 | 1.00 | 3.80 | 8.83 | 8.48 | 3.31 | -0.03 | **92.28** |
| 1495 | 96.81 | **98.40** | 72.74 | 82.80 | 96.81 | 91.28 | 83.96 | 77.51 | 47.11 | 5.10 | 7.35 |
| 1499 | 77.33 | 62.99 | 73.68 | 75.28 | 69.99 | 78.48 | 67.82 | 29.60 | 5.82 | 24.93 | **82.40** |
| 1510 | 67.07 | **78.02** | 63.73 | 59.77 | 71.32 | 69.84 | 70.55 | 69.47 | 54.59 | 63.57 | 74.26 |
| 1523 | 21.16 | **41.21** | 25.48 | 24.57 | 28.16 | 23.05 | 31.35 | 18.50 | 2.45 | 21.04 | 19.60 |
| 4153 | 55.78 | 58.03 | 59.15 | 46.23 | 56.41 | 58.44 | 50.44 | 39.43 | 27.80 | 36.21 | **62.29** |
| 40496 | **54.02** | 34.79 | 42.35 | 37.51 | 49.74 | 42.35 | 27.41 | 37.33 | 31.80 | 39.78 | 32.25 |
| 40682 | 58.32 | 86.29 | 57.89 | 59.22 | 57.39 | 86.05 | 17.95 | 40.58 | **86.94** | 17.86 | 53.15 |
| 40705 | 43.49 | 37.92 | 0.73 | 30.63 | **47.90** | 46.28 | 31.38 | 10.71 | -4.40 | 32.90 | 45.45 |
| 42261 | 62.01 | **90.39** | 60.13 | 58.15 | 59.78 | 56.78 | 55.10 | 49.09 | 77.20 | 52.53 | 85.15 |
| 42585 | 60.84 | 30.52 | 51.26 | 60.71 | 91.79 | 43.40 | 72.10 | 38.30 | 23.04 | 3.50 | **95.05** |
| Mean | 55.54 | 48.49 | 51.43 | 53.56 | 55.93 | 54.57 | 52.28 | 40.37 | 31.32 | 26.95 | **57.43** |
| Mean-Rank | 4.69 | 5.65 | 5.72 | 5.24 | 4.69 | 5.01 | 5.62 | 8.18 | 8.22 | 8.85 | **4.13** |
| Top-3 | 11 | 12 | 6 | 5 | 15 | 11 | 9 | 1 | 6 | 4 | **21** |
| Top-1 | 4 | 7 | 3 | 0 | 2 | 1 | 1 | 0 | 2 | 0 | **12** |

Table 10: Evaluation of clustering quality with the ARI metric on Synthetic Gaussian datasets (higher scores reflect better performance).

| ID | KM | GMM | AE-KM | AE-GMM | DEC | IDEC | IDC | G-CEALS | TabPFN | SCARF | ZEUS |
|---|---|---|---|---|---|---|---|---|---|---|---|
| 0 | 86.16 | 82.13 | 83.44 | 79.66 | 87.06 | 77.26 | 60.64 | 58.57 | 65.42 | 4.06 | **87.65** |
| 1 | **88.66** | 87.36 | 81.13 | 79.61 | 85.84 | 75.62 | 44.44 | 54.10 | 40.26 | 4.21 | 72.04 |
| 2 | 81.30 | 26.46 | 71.57 | 78.04 | **86.65** | 78.91 | 53.53 | 48.59 | 20.42 | 11.44 | 82.96 |
| 3 | 95.02 | 36.77 | 92.03 | 76.44 | 92.74 | 89.81 | 65.03 | 62.67 | 50.57 | 22.29 | **95.76** |
| 4 | 90.64 | **92.97** | 82.65 | 77.80 | 91.61 | 84.51 | 77.56 | 69.70 | 61.65 | 8.80 | 92.85 |
| 5 | 89.83 | 25.85 | 86.52 | 69.06 | 90.01 | 88.31 | 70.38 | 67.70 | 20.51 | 10.31 | 90.72 |
| 6 | 96.21 | 98.07 | 85.99 | 94.26 | 94.24 | 92.39 | 80.18 | 73.15 | 86.50 | 6.51 | 97.89 |
| 7 | 92.59 | 30.52 | 81.73 | 79.06 | 91.35 | 89.80 | 46.44 | 58.10 | 26.62 | 10.14 | **93.60** |
| 8 | 88.29 | **89.62** | 81.15 | 76.53 | 87.56 | 78.37 | 72.72 | 50.30 | 68.54 | 13.06 | 84.00 |
| 9 | 93.36 | **98.54** | 68.58 | 85.42 | 91.14 | 85.29 | 76.64 | 56.89 | 65.02 | 11.37 | 94.55 |
| 10 | 79.26 | **93.32** | 72.75 | 90.90 | 72.54 | 73.51 | 77.64 | 48.47 | 89.15 | -0.04 | 87.32 |
| 11 | 93.60 | 98.48 | 90.77 | 89.11 | 95.03 | 92.19 | 74.56 | 64.24 | 68.28 | 2.11 | 97.45 |
| 12 | **86.82** | 77.45 | 73.64 | 71.40 | 86.04 | 77.93 | 60.55 | 58.45 | 55.27 | 19.11 | 86.04 |
| 13 | 96.77 | **98.07** | 94.05 | 90.15 | 96.11 | 85.66 | 86.14 | 78.20 | 80.24 | 5.29 | 96.17 |
| 14 | 90.73 | 43.20 | 79.65 | 69.09 | 92.68 | 77.93 | 72.80 | 67.84 | 12.93 | 8.16 | **94.60** |
| 15 | 93.58 | 96.23 | 85.74 | 89.30 | 94.43 | 89.41 | 72.54 | 72.88 | 91.02 | 3.24 | **96.68** |
| 16 | **94.70** | 92.45 | 90.98 | 89.41 | 94.53 | 88.14 | 77.56 | 85.86 | 59.58 | 4.85 | 87.19 |
| 17 | 85.26 | **87.10** | 62.23 | 79.52 | 84.08 | 75.00 | 50.45 | 42.55 | 50.06 | 7.69 | 86.29 |
| 18 | 88.77 | **94.83** | 84.47 | 84.74 | 87.39 | 83.46 | 61.36 | 67.96 | 74.90 | 7.69 | 83.01 |
| 19 | 86.54 | **89.11** | 76.18 | 78.46 | 86.06 | 67.99 | 47.46 | 70.50 | 32.39 | 6.02 | 73.85 |
| Mean | **89.90** | 76.93 | 81.26 | 81.40 | 89.35 | 82.57 | 66.43 | 62.84 | 55.97 | 8.32 | 89.03 |
| Mean-Rank | **2.65** | 3.65 | 5.65 | 5.60 | 3.23 | 5.70 | 7.95 | 8.90 | 8.75 | 11.00 | 2.92 |
| Top-3 | **16** | 13 | 0 | 1 | 15 | 0 | 0 | 0 | 1 | 0 | 14 |
| Top-1 | 3 | **10** | 0 | 0 | 1 | 0 | 0 | 0 | 0 | 0 | 6 |

Table 11: Assessment of clustering quality based on the ARI metric on Synthetic Transformed datasets (higher score indicate better performance).

| ID | KM | GMM | AE-KM | AE-GMM | DEC | IDEC | IDC | G-CEALS | TabPFN | SCARF | ZEUS |
|---|---|---|---|---|---|---|---|---|---|---|---|
| 0 | 60.56 | 79.33 | 46.31 | 68.98 | 76.93 | 46.35 | 56.67 | 29.42 | 6.36 | 0.94 | **87.42** |
| 1 | 93.30 | **96.55** | 76.57 | 83.99 | 91.29 | 70.91 | 66.59 | 73.09 | 16.59 | 1.22 | 89.40 |
| 2 | 89.76 | **96.31** | 57.79 | 65.99 | 94.42 | 66.34 | 72.25 | 48.34 | 10.00 | 1.10 | 95.88 |
| 3 | 70.16 | 50.38 | 56.23 | 76.06 | **83.06** | 60.09 | 65.13 | 54.54 | 15.36 | 1.99 | 80.98 |
| 4 | 44.32 | **80.63** | 47.60 | 44.77 | 68.39 | 50.12 | 36.00 | 49.37 | 19.14 | 1.48 | 73.66 |
| 5 | 91.49 | **96.51** | 74.48 | 89.11 | 86.46 | 77.84 | 60.00 | 66.47 | 15.12 | 1.25 | 81.63 |
| 6 | 60.98 | 81.63 | 61.97 | 76.29 | 75.58 | 61.89 | 69.43 | 69.84 | 18.83 | 2.11 | **87.02** |
| 7 | 76.80 | 83.98 | 71.37 | 77.69 | 86.56 | 77.49 | 76.75 | 56.51 | 18.94 | 3.20 | **89.57** |
| 8 | 89.77 | 77.10 | 85.53 | 84.42 | 93.33 | 92.96 | 90.69 | 78.87 | 24.26 | 5.91 | **98.11** |
| 9 | 84.40 | **98.48** | 82.64 | 93.03 | 97.14 | 81.02 | 91.02 | 65.62 | 9.01 | 1.66 | 97.20 |
| 10 | 82.47 | **91.91** | 61.00 | 86.16 | 87.58 | 46.08 | 70.42 | 39.27 | 14.11 | 0.46 | 86.42 |
| 11 | 90.77 | **99.32** | 81.41 | 91.18 | 94.78 | 88.12 | 80.75 | 70.29 | 18.31 | 1.72 | 93.19 |
| 12 | 67.64 | **91.80** | 52.87 | 63.56 | 85.45 | 60.61 | 73.66 | 55.59 | 28.50 | 3.21 | 87.75 |
| 13 | 85.99 | 50.80 | 45.48 | 35.34 | 29.04 | 12.91 | 30.22 | 32.89 | 35.78 | 6.01 | **91.60** |
| 14 | 93.52 | 54.56 | 15.81 | 22.79 | 94.12 | 38.21 | 68.54 | 15.10 | 15.08 | 5.54 | **98.49** |
| 15 | 36.62 | 30.93 | 37.71 | **43.70** | 42.78 | 40.65 | 32.64 | 29.70 | 16.68 | 0.63 | 28.96 |
| 16 | 95.62 | 65.11 | 90.13 | 94.97 | 97.47 | 95.32 | 84.79 | 98.27 | 17.27 | 9.58 | **99.01** |
| 17 | 60.54 | 95.74 | 56.24 | 91.71 | 78.72 | 59.11 | 82.54 | 13.29 | 7.33 | -0.10 | **96.44** |
| 18 | 63.92 | 17.39 | 58.47 | 70.21 | 54.61 | 47.93 | 64.15 | 10.01 | 3.79 | 0.65 | **79.70** |
| 19 | 62.12 | 79.06 | 49.36 | 65.77 | 81.07 | 51.19 | 63.41 | 26.84 | 2.73 | 1.06 | **84.13** |
| Mean | 75.04 | 75.88 | 60.45 | 71.29 | 79.94 | 61.26 | 66.78 | 49.17 | 15.66 | 2.48 | **86.33** |
| Mean-Rank | 4.80 | 3.50 | 6.85 | 4.50 | 3.20 | 6.35 | 6.05 | 7.80 | 9.75 | 11.00 | **2.20** |
| Top-3 | 4 | 14 | 0 | 6 | 15 | 2 | 1 | 1 | 0 | 0 | **17** |
| Top-1 | 0 | 8 | 0 | 1 | 1 | 0 | 0 | 0 | 0 | 0 | **10** |

## F.2 Are ZEUS's assignments well calibrated?

Tables 12, 13 and 14 contain detailed extension of Table 3 from the main part of the paper. Similar to the Tables in Appendix F.1, these also include additional rows: Mean, Mean-Rank, Top-3, and Top-1, which aggregate the results presented in each table.

ZEUS consistently attains at least the second position across all statistics in every table presented in this section. For the Synthetic Transformed datasets, it is undeniably the best. However, for the OpenML datasets, it is outperformed by IDEC, and for the Synthetic Gaussian datasets, GMM takes the lead in the Top-1 metric, while for the other statistics, k-means performs better.

Table 12: Soft clustering performance of ZEUS versus competing methods on real-world datasets, measured by Brier score (lower is better).

| ID | KM | GMM | AE-KM | AE-GMM | DEC | IDEC | G-CEALS | ZEUS |
|---|---|---|---|---|---|---|---|---|
| 14 | 0.7660 | 0.8579 | 0.8240 | 0.7144 | **0.5338** | 0.6409 | 0.7993 | 0.7006 |
| 15 | 0.0858 | 0.1548 | 0.2100 | 0.1791 | 0.1489 | **0.0644** | 0.2003 | 0.0973 |
| 16 | 0.5012 | 0.4982 | 0.4224 | 0.5372 | 0.3726 | 0.5020 | 0.6099 | **0.2761** |
| 18 | 0.6060 | 0.8529 | 0.7312 | 0.6961 | 0.6099 | **0.5354** | 0.6855 | 0.7094 |
| 22 | 0.8316 | 0.7533 | 1.0686 | 0.7684 | **0.5071** | 0.6496 | 0.7201 | 0.7173 |
| 35 | **0.0710** | 0.5016 | 0.4678 | 0.5711 | 0.5881 | 0.3987 | 0.3836 | 0.2552 |
| 51 | 0.3741 | 0.3741 | 0.3687 | 0.3844 | **0.3297** | 0.3345 | 0.3997 | 0.3333 |
| 53 | 0.4074 | 0.7052 | 0.3556 | 0.6285 | 0.3361 | **0.2979** | 0.5234 | 0.4000 |
| 56 | 0.2391 | 0.2344 | 0.2520 | 0.2703 | 0.2147 | 0.2088 | 0.2517 | **0.1829** |
| 61 | 0.2267 | **0.0570** | 0.3600 | 0.3796 | 0.3881 | 0.3445 | 0.4167 | 0.0979 |
| 187 | 0.0899 | **0.0622** | 0.1349 | 0.1146 | 0.3380 | 0.0785 | 0.2518 | 0.0762 |
| 377 | 0.8633 | 0.8320 | 0.6200 | 0.6275 | 0.5316 | **0.5047** | 0.5739 | 0.8508 |
| 458 | 0.0285 | 0.0214 | 0.5222 | 0.3407 | 0.3203 | 0.3656 | 0.4104 | **0.0048** |
| 481 | 0.8038 | 0.8134 | 0.4536 | **0.2136** | 0.3111 | 0.2146 | 0.4422 | 0.6986 |
| 694 | 1.0155 | 0.8831 | 0.9858 | 0.9342 | 0.7616 | 0.7525 | **0.7038** | 0.7621 |
| 721 | **0.3400** | **0.3400** | 0.6620 | 0.6031 | 0.4491 | 0.4359 | 0.7937 | **0.3400** |
| 733 | 0.1722 | 0.2859 | 0.3082 | 0.4937 | 0.2115 | **0.1129** | 0.3195 | 0.1244 |
| 745 | 0.2516 | 0.2516 | **0.1359** | 0.2766 | 0.3016 | 0.1793 | 0.2388 | 0.1384 |
| 756 | 0.2767 | 0.3615 | 0.2113 | 0.3262 | 0.3178 | **0.1329** | 0.4134 | 0.2767 |
| 796 | **0.0574** | 0.7943 | 0.1340 | 0.3088 | 0.2046 | 0.3002 | 0.5269 | 0.6253 |
| 820 | 0.2979 | 0.3720 | 0.3557 | 0.2876 | **0.2334** | 0.2594 | 0.3981 | 0.4596 |
| 840 | 0.8780 | 0.2790 | 0.2868 | 0.2625 | 0.2746 | **0.2445** | 0.4642 | 0.4665 |
| 854 | 0.6329 | 0.8203 | 0.3468 | 0.4481 | 0.4469 | 0.3372 | 0.3678 | **0.1266** |
| 1462 | 0.8484 | 0.8978 | 0.7688 | 0.8310 | 0.6674 | 0.7714 | 0.6747 | **0.0394** |
| 1495 | 0.0160 | **0.0080** | 0.1696 | 0.0397 | 0.2667 | 0.0713 | 0.1583 | 0.7200 |
| 1499 | 0.2190 | 0.2933 | 0.1867 | 0.1961 | 0.3939 | 0.1376 | 0.4063 | **0.1239** |
| 1510 | 0.1441 | **0.1161** | 0.2257 | 0.1954 | 0.2195 | 0.1432 | 0.2060 | 0.1371 |
| 1523 | 1.0516 | 0.7284 | 0.9019 | 0.9708 | **0.6099** | 0.7782 | 0.6316 | 1.0002 |
| 4153 | 0.7778 | 0.7178 | 0.6311 | 0.8111 | 0.5757 | 0.6377 | 0.8360 | **0.4651** |
| 40496 | **0.5064** | 0.9563 | 0.8312 | 0.8432 | 0.6733 | 0.6143 | 0.7376 | 1.0124 |
| 40682 | 0.2233 | 0.0747 | 0.2456 | 0.2251 | 0.2694 | **0.0685** | 0.3210 | 0.2884 |
| 40705 | 0.3879 | 0.3837 | 0.7095 | 0.4415 | **0.2541** | 0.3347 | 0.4304 | 0.3254 |
| 42261 | 0.2267 | **0.0570** | 0.3413 | 0.3760 | 0.3889 | 0.2854 | 0.3800 | 0.1081 |
| 42585 | 0.6279 | 0.9790 | 0.5558 | 0.6111 | 0.3494 | 0.7425 | 0.3795 | **0.0383** |
| Mean | 0.4366 | 0.4799 | 0.4643 | 0.4679 | 0.3941 | **0.3671** | 0.4722 | 0.3817 |
| Mean-Rank | 4.54 | 4.94 | 5.12 | 5.50 | 3.88 | **2.85** | 5.65 | 3.51 |
| Top-3 | 12 | 9 | 8 | 4 | 19 | **22** | 6 | 21 |
| Top-1 | 3 | 5 | 1 | 1 | 6 | **8** | 1 | **8** |

Table 13: Soft clustering performance (Brier score) of ZEUS compared to baseline methods on Synthetic Gaussian datasets (lower score reflect better quality).

| ID | KM | GMM | AE-KM | AE-GMM | DEC | IDEC | G-CEALS | ZEUS |
|---|---|---|---|---|---|---|---|---|
| 0 | 0.1448 | 0.2200 | 0.1741 | 0.3558 | 0.4251 | 0.3264 | 0.3921 | **0.1242** |
| 1 | **0.1218** | 0.1886 | 0.2337 | 0.4718 | 0.4379 | 0.4109 | 0.4377 | 0.4502 |
| 2 | 0.1985 | 0.9985 | 0.2995 | **0.1922** | 0.4303 | 0.2321 | 0.5400 | 0.1952 |
| 3 | 0.0528 | 0.9078 | 0.0927 | 0.3012 | 0.6196 | 0.1628 | 0.4194 | **0.0413** |
| 4 | 0.1005 | **0.0575** | 0.1793 | 0.3351 | 0.4955 | 0.2312 | 0.3886 | 0.0792 |
| 5 | 0.1020 | 1.0950 | 0.1333 | 0.4892 | 0.3355 | 0.2059 | 0.3995 | **0.0881** |
| 6 | 0.0350 | **0.0156** | 0.1200 | 0.0489 | 0.2971 | 0.0770 | 0.3561 | 0.0209 |
| 7 | 0.0632 | 0.9781 | 0.1599 | 0.2610 | 0.4693 | 0.1207 | 0.6146 | **0.0543** |
| 8 | 0.1168 | **0.0844** | 0.2079 | 0.2786 | 0.4626 | 0.2970 | 0.5975 | 0.1631 |
| 9 | 0.0677 | **0.0092** | 0.3552 | 0.1925 | 0.2167 | 0.2090 | 0.4586 | 0.0573 |
| 10 | 0.1091 | **0.0292** | 0.1469 | 0.0460 | 0.3300 | 0.1229 | 0.3127 | 0.0655 |
| 11 | 0.0558 | **0.0117** | 0.0791 | 0.1370 | 0.4110 | 0.1122 | 0.4437 | 0.0234 |
| 12 | **0.1361** | 0.2296 | 0.4022 | 0.3492 | 0.5917 | 0.3230 | 0.3583 | 0.1545 |
| 13 | 0.0216 | **0.0115** | 0.0356 | 0.0629 | 0.3223 | 0.0980 | 0.1947 | 0.0270 |
| 14 | 0.1048 | 0.8000 | 0.3009 | 0.3776 | 0.3359 | 0.2380 | 0.3818 | **0.0626** |
| 15 | 0.0734 | **0.0278** | 0.2574 | 0.0746 | 0.3458 | 0.2483 | 0.3120 | 0.0352 |
| 16 | 0.0537 | **0.0448** | 0.0958 | 0.1747 | 0.1640 | 0.2528 | 0.1454 | 0.1770 |
| 17 | 0.1309 | 0.3513 | 0.4178 | 0.2891 | 0.4313 | 0.2756 | 0.5425 | **0.1308** |
| 18 | 0.1035 | **0.0452** | 0.1458 | 0.2319 | 0.5193 | 0.2117 | 0.2961 | 0.1939 |
| 19 | 0.1475 | **0.1132** | 0.3094 | 0.2989 | 0.2444 | 0.4612 | 0.3015 | 0.3942 |
| Mean | **0.0970** | 0.3110 | 0.2073 | 0.2484 | 0.3943 | 0.2308 | 0.3946 | 0.1269 |
| Mean-Rank | **2.30** | 3.25 | 4.55 | 4.95 | 6.65 | 4.85 | 6.75 | 2.70 |
| Top-3 | **19** | 13 | 6 | 2 | 1 | 3 | 0 | 16 |
| Top-1 | 2 | **11** | 0 | 1 | 0 | 0 | 0 | 6 |

Table 14: Evaluation of soft clustering quality on Synthetic Transformed datasets using Brier score (lower score indicate better performance).

| ID | KM | GMM | AE-KM | AE-GMM | DEC | IDEC | G-CEALS | ZEUS |
|---|---|---|---|---|---|---|---|---|
| 0 | 0.5116 | 0.2135 | 0.6639 | 0.4457 | 0.5911 | 0.5063 | 0.7116 | **0.1282** |
| 1 | 0.0666 | **0.0484** | 0.2328 | 0.1514 | 0.4755 | 0.5045 | 0.2806 | 0.2718 |
| 2 | 0.0883 | **0.0330** | 0.4996 | 0.2822 | 0.4424 | 0.3441 | 0.5990 | 0.0394 |
| 3 | 0.3254 | 0.6286 | 0.5872 | **0.2118** | 0.3241 | 0.4862 | 0.7537 | 0.2678 |
| 4 | 0.6757 | **0.1547** | 0.5580 | 0.6123 | 0.3920 | 0.5158 | 0.3804 | 0.3190 |
| 5 | 0.0887 | **0.0359** | 0.3201 | 0.1517 | 0.6415 | 0.4114 | 0.3869 | 0.2802 |
| 6 | 0.5771 | 0.1788 | 0.4789 | 0.4049 | 0.5718 | 0.3069 | 0.5448 | **0.1763** |
| 7 | 0.3634 | 0.2875 | 0.4281 | 0.3326 | 0.5478 | 0.3379 | 0.6068 | **0.2461** |
| 8 | 0.1048 | 0.2738 | 0.1271 | 0.0817 | 0.4517 | 0.0784 | 0.3278 | **0.0167** |
| 9 | 0.3014 | **0.0120** | 0.3120 | 0.0940 | 0.4474 | 0.3329 | 0.2284 | 0.0481 |
| 10 | 0.1785 | **0.0763** | 0.3674 | 0.1249 | 0.3845 | 0.5057 | 0.5420 | 0.1387 |
| 11 | 0.1053 | **0.0130** | 0.1684 | 0.1128 | 0.6071 | 0.2580 | 0.3306 | 0.1768 |
| 12 | 0.3682 | **0.0657** | 0.6462 | 0.4740 | 0.3838 | 0.4585 | 0.6318 | 0.1936 |
| 13 | 0.0604 | 0.2549 | 0.4206 | 0.4292 | 0.3711 | 0.4073 | 0.3436 | **0.0355** |
| 14 | 0.0445 | 0.7982 | 1.1164 | 0.6259 | 0.3672 | 0.4949 | 0.6715 | **0.0097** |
| 15 | 0.7142 | 0.7088 | 0.6256 | 0.7924 | 0.5801 | 0.5741 | **0.4807** | 0.8854 |
| 16 | 0.0524 | 0.4745 | 0.1194 | **0.0303** | 0.5832 | 0.1743 | 0.1014 | 0.0809 |
| 17 | 0.2586 | 0.0230 | 0.2616 | 0.0394 | 0.3480 | 0.2156 | 0.6361 | **0.0177** |
| 18 | 0.2081 | 0.6698 | 0.3781 | 0.2409 | 0.3774 | 0.3861 | 0.7879 | **0.1056** |
| 19 | 0.5120 | 0.1821 | 0.7577 | 0.6415 | 0.3892 | 0.4845 | 0.5570 | **0.1542** |
| Mean | 0.2803 | 0.2566 | 0.4535 | 0.3140 | 0.4638 | 0.3892 | 0.4951 | **0.1796** |
| Mean-Rank | 3.95 | 3.00 | 5.85 | 3.95 | 5.75 | 5.00 | 6.15 | **2.35** |
| Top-3 | 9 | 14 | 0 | 12 | 4 | 3 | 2 | **16** |
| Top-1 | 0 | 8 | 0 | 2 | 0 | 0 | 1 | **9** |

### F.3 How helpful is regularization for ZEUS?

Tables 15, 16 and 17 provide extended versions of Table 4 , which analyzes the impact of different combinations of regularization functions. Their structure is analogous to the other tables presented in Appendix F.

The conclusions that can be drawn from these extended tables, along with their statistical summaries, are consistent with our original claims. The model with both $\mathcal{L}_{sep}$ and $\mathcal{L}_{cp}$ regularizers performs best on real-world data collections and consistently ranks at least second in the considered statistics for synthetic data. On these generated datasets, it is frequently outperformed by the variant that includes the $\mathcal{L}_{cp}$ component alone. By contrast, the remaining two approaches clearly grapple with proper clustering of the synthetic datasets.

Table 15: Impact of regularization components on OpenML datasets, evaluated using the ARI score (higher is better).

| ID | $\mathcal{L}_{prob}$ | $\mathcal{L}_{prob} + \mathcal{L}_{sep}$ | $\mathcal{L}_{prob} + \mathcal{L}_{cp}$ | $\mathcal{L}_{prob} + \mathcal{L}_{sep} + \mathcal{L}_{cp}$ |
|---|---|---|---|---|
| 14 | 48.40 | 43.49 | 46.44 | **50.56** |
| 15 | 86.64 | **87.17** | 82.94 | 81.28 |
| 16 | 69.87 | 73.33 | **81.87** | 74.03 |
| 18 | 43.06 | 50.80 | 49.62 | **51.63** |
| 22 | 63.27 | 56.86 | **65.58** | 56.05 |
| 35 | 52.33 | 74.58 | 76.90 | **85.12** |
| 51 | 32.31 | 31.54 | 41.84 | **43.66** |
| 53 | 5.17 | 27.38 | 25.85 | **35.76** |
| 56 | 58.48 | 61.34 | 20.84 | **66.41** |
| 61 | 48.93 | **88.57** | 65.37 | 85.15 |
| 187 | 88.38 | **91.50** | 88.22 | 88.19 |
| 377 | 46.31 | 45.16 | 54.56 | **55.30** |
| 458 | 98.16 | 98.49 | 98.49 | **99.19** |
| 481 | 2.77 | **25.87** | 1.53 | 8.68 |
| 694 | 42.76 | **48.86** | 30.35 | 33.38 |
| 721 | **43.28** | 5.39 | **43.28** | **43.28** |
| 733 | 34.35 | 24.94 | **76.83** | 74.97 |
| 745 | 59.56 | 22.80 | 4.46 | **73.85** |
| 756 | **76.15** | 63.49 | 38.42 | 50.31 |
| 796 | 6.56 | 3.96 | **14.52** | 13.99 |
| 820 | **39.95** | **39.95** | 34.72 | 28.87 |
| 840 | 0.95 | 4.13 | **31.14** | 27.66 |
| 854 | 12.37 | 12.37 | 24.32 | **76.14** |
| 1462 | 92.00 | **94.25** | 62.85 | 92.28 |
| 1495 | -0.29 | **98.40** | 7.35 | 7.35 |
| 1499 | 60.63 | 44.86 | 71.34 | **82.40** |
| 1510 | 70.09 | 73.67 | 72.51 | **74.26** |
| 1523 | 25.68 | **28.13** | 17.38 | 19.60 |
| 4153 | 38.28 | 51.26 | 50.24 | **62.29** |
| 40496 | **34.15** | 33.63 | 30.68 | 32.25 |
| 40682 | -0.39 | 23.38 | 42.79 | **53.15** |
| 40705 | 40.00 | 40.51 | 38.18 | **45.45** |
| 42261 | 48.93 | **88.57** | 65.37 | 85.15 |
| 42585 | 53.97 | 95.82 | **97.39** | 95.05 |
| Mean | 44.80 | 51.60 | 48.65 | **57.43** |
| Mean-rank | 3.00 | 2.37 | 2.68 | **1.96** |
| Top-3 | 20 | 26 | 25 | **30** |
| Top-1 | 2 | 9 | 6 | **15** |

Table 16: Effect of regularization components on Synthetic Gaussian datasets, assessed by the ARI metric (higher score indicate better quality).

| ID | $\mathcal{L}_{prob}$ | $\mathcal{L}_{prob} + \mathcal{L}_{sep}$ | $\mathcal{L}_{prob} + \mathcal{L}_{cp}$ | $\mathcal{L}_{prob} + \mathcal{L}_{sep} + \mathcal{L}_{cp}$ |
|---|---|---|---|---|
| 0 | 72.89 | 68.81 | 85.85 | **87.65** |
| 1 | **89.50** | 71.68 | 88.72 | 72.04 |
| 2 | 80.18 | 63.23 | **92.59** | 82.96 |
| 3 | 77.05 | 79.22 | 91.23 | **95.76** |
| 4 | 72.32 | 84.96 | 92.82 | **92.85** |
| 5 | **91.88** | 88.98 | 82.25 | 90.72 |
| 6 | 81.85 | 81.18 | 87.49 | **97.89** |
| 7 | 85.33 | 79.93 | **93.75** | 93.60 |
| 8 | 72.54 | 76.12 | **87.92** | 84.00 |
| 9 | 95.47 | **95.97** | 93.59 | 94.55 |
| 10 | 88.23 | **89.61** | 85.07 | 87.32 |
| 11 | 97.26 | 94.91 | 96.55 | **97.45** |
| 12 | 77.58 | 78.07 | 85.81 | **86.04** |
| 13 | 74.41 | 76.08 | 96.07 | **96.17** |
| 14 | 84.00 | 82.11 | **95.39** | 94.60 |
| 15 | 77.65 | 78.94 | **97.10** | 96.68 |
| 16 | 93.82 | 93.46 | **95.82** | 87.19 |
| 17 | **88.83** | 86.38 | 88.05 | 86.29 |
| 18 | 89.90 | **90.21** | 90.02 | 83.01 |
| 19 | 76.62 | 77.69 | **85.63** | 73.85 |
| Mean | 83.37 | 81.88 | **90.59** | 89.03 |
| Mean-rank | 2.80 | 3.00 | **2.00** | 2.20 |
| Top-3 | 14 | 13 | **17** | 16 |
| Top-1 | 3 | 3 | **7** | **7** |

Table 17: Regularisation impact on Synthetic Transformed datasets, measured by ARI (higher is better).

| ID | $\mathcal{L}_{prob}$ | $\mathcal{L}_{prob} + \mathcal{L}_{sep}$ | $\mathcal{L}_{prob} + \mathcal{L}_{cp}$ | $\mathcal{L}_{prob} + \mathcal{L}_{sep} + \mathcal{L}_{cp}$ |
|---|---|---|---|---|
| 0 | 67.29 | 71.62 | 85.76 | **87.42** |
| 1 | 91.98 | 91.46 | **96.88** | 89.40 |
| 2 | 93.17 | 93.84 | 92.36 | **95.88** |
| 3 | 68.26 | 71.68 | 80.53 | **80.98** |
| 4 | 80.93 | 81.64 | **84.22** | 73.66 |
| 5 | 81.38 | 80.06 | **95.99** | 81.63 |
| 6 | 84.29 | 82.06 | 80.67 | **87.02** |
| 7 | 83.35 | 73.07 | **96.83** | 89.57 |
| 8 | 97.42 | **98.61** | 97.16 | 98.11 |
| 9 | **97.63** | 96.43 | 96.37 | 97.20 |
| 10 | **91.32** | 77.37 | 90.58 | 86.42 |
| 11 | 91.84 | 89.07 | **99.17** | 93.19 |
| 12 | 74.93 | **92.20** | 91.14 | 87.75 |
| 13 | 0.32 | 0.73 | 90.67 | **91.60** |
| 14 | 98.00 | **98.83** | 98.16 | 98.49 |
| 15 | 43.28 | **48.54** | 30.50 | 28.96 |
| 16 | 93.34 | 90.38 | 98.11 | **99.01** |
| 17 | 97.62 | **98.21** | 95.86 | 96.44 |
| 18 | 82.64 | 82.22 | **86.94** | 79.70 |
| 19 | 77.93 | 67.71 | 83.78 | **84.13** |
| Mean | 79.85 | 79.29 | **88.58** | 86.33 |
| Mean-rank | 2.80 | 2.70 | 2.30 | **2.20** |
| Top-3 | 15 | 14 | 15 | **16** |
| Top-1 | 2 | 5 | 6 | **7** |

## F.4 What data-generating prior is optimal?

Tables 18, 19, and 20 expand upon the findings presented in Table 5 by examining the influence of various data-generating priors across all datasets within each group. Their format is consistent with other tables in Appendix F.

Table 18: Evaluation of clustering quality with the ARI metric on real-world datasets for different combinations of data-generating probabilistic models (higher scores reflect better performance).

| ID | Gauss. + Cat. | NN-transf. + Cat. | Gauss. + NN-transf. | Gauss. + NN-transf. + Cat. |
|---|---|---|---|---|
| 14 | 36.81 | _49.84_ | 49.84 | **50.56** |
| 15 | _85.52_ | **85.57** | 83.46 | 81.28 |
| 16 | 57.15 | 73.05 | **76.01** | _74.03_ |
| 18 | 42.83 | 46.48 | **55.29** | _51.63_ |
| 22 | 46.45 | **62.23** | _60.63_ | 56.02 |
| 35 | **87.48** | 76.48 | 84.25 | _85.12_ |
| 51 | -0.52 | 38.26 | _39.76_ | **43.66** |
| 53 | **37.57** | 29.79 | 27.38 | _35.76_ |
| 56 | 48.40 | 56.35 | _59.92_ | **66.41** |
| 61 | 68.44 | 62.26 | _85.08_ | **85.15** |
| 187 | _89.77_ | **94.87** | 88.22 | 88.19 |
| 377 | 35.12 | _58.34_ | **58.95** | 55.30 |
| 458 | 47.43 | _98.79_ | 98.47 | **99.19** |
| 481 | 6.46 | _6.94_ | 3.58 | **8.68** |
| 694 | _27.75_ | 19.90 | 27.69 | **33.38** |
| 721 | **43.28** | _43.28_ | 38.13 | 43.28 |
| 733 | 73.48 | -7.69 | **75.77** | _74.97_ |
| 745 | 4.46 | _67.51_ | 28.11 | **73.85** |
| 756 | 4.46 | **59.53** | 4.46 | _50.31_ |
| 796 | _14.80_ | 5.94 | **15.29** | 13.99 |
| 820 | 19.26 | **36.77** | _33.71_ | 28.87 |
| 840 | -0.83 | 0.69 | _2.57_ | **27.66** |
| 854 | 24.32 | _73.92_ | 12.56 | **76.14** |
| 1462 | 1.92 | **92.56** | _92.56_ | 92.28 |
| 1495 | -0.29 | 0.03 | **55.17** | _7.35_ |
| 1499 | 58.62 | _74.43_ | 71.07 | **82.40** |
| 1510 | 72.50 | _72.95_ | 70.08 | **74.26** |
| 1523 | _20.38_ | 18.69 | **22.01** | 19.60 |
| 4153 | 41.49 | 46.33 | _56.13_ | **62.29** |
| 40496 | **33.20** | 31.69 | 30.61 | _32.25_ |
| 40682 | _64.49_ | 58.59 | **77.21** | 53.15 |
| 40705 | 29.52 | _39.74_ | 37.15 | **45.45** |
| 42261 | 66.34 | 62.26 | _81.76_ | **85.15** |
| 42585 | 92.06 | _94.29_ | 64.95 | **95.05** |
| Mean | 40.59 | 50.90 | _52.00_ | **57.43** |
| Mean-Rank | 3.10 | 2.56 | _2.46_ | **1.88** |
| Top-3 | 17 | _27_ | 26 | **31** |
| Top-1 | 3 | 5 | _8_ | **16** |

Table 19: Effect of prior selection on ZEUS's ARI performance on Synthetic Gaussian datasets (higher scores indicate better performance).

| ID | Gauss. + Cat. | NN-transf. + Cat. | Gauss. + NN-transf. | Gauss. + NN-transf. + Cat. |
|----|----|----|----|----|
| 0 | **88.78** | 87.09 | 82.89 | 87.65 |
| 1 | **87.74** | 79.51 | 80.38 | 72.04 |
| 2 | **91.30** | 87.30 | 20.50 | 82.96 |
| 3 | 95.08 | 93.27 | 44.56 | **95.76** |
| 4 | 94.40 | **94.46** | 92.48 | 92.85 |
| 5 | **91.43** | 90.08 | 30.45 | 90.72 |
| 6 | **98.05** | 97.22 | 98.02 | 97.89 |
| 7 | **94.56** | 94.43 | 26.90 | 93.60 |
| 8 | 87.65 | 87.95 | **88.05** | 84.00 |
| 9 | **95.92** | 92.70 | 94.79 | 94.55 |
| 10 | **90.99** | 86.78 | 90.07 | 87.32 |
| 11 | 96.86 | 96.52 | 96.74 | **97.45** |
| 12 | **88.21** | 86.25 | 85.55 | 86.04 |
| 13 | 96.07 | 93.39 | 93.81 | **96.17** |
| 14 | **95.42** | 92.12 | 32.24 | 94.60 |
| 15 | **97.71** | 95.84 | 97.14 | 96.68 |
| 16 | 95.24 | 94.49 | **95.50** | 87.19 |
| 17 | **89.13** | 84.54 | 86.66 | 86.29 |
| 18 | **90.14** | 87.66 | 89.37 | 83.01 |
| 19 | **87.43** | 76.30 | 79.00 | 73.85 |
| Mean | **92.61** | 89.90 | 75.25 | 89.03 |
| Mean-Rank | **1.35** | 3.05 | 2.80 | 2.80 |
| Top-3 | **20** | 13 | 12 | 15 |
| Top-1 | **14** | 1 | 2 | 3 |

Table 20: Influence of the chosen data-generating prior on ZEUS performance over Synthetic Transformed datasets, reported in terms of ARI (larger values denote better clustering quality).

| ID | Gauss. + Cat. | NN-transf. + Cat. | Gauss. + NN-transf. | Gauss. + NN-transf. + Cat. |
|----|----|----|----|----|
| 0 | 61.54 | 87.33 | **88.33** | 87.42 |
| 1 | 84.60 | **95.73** | 94.60 | 89.40 |
| 2 | 83.73 | 93.89 | **97.59** | 95.88 |
| 3 | 45.52 | 75.48 | 52.79 | **80.98** |
| 4 | 76.52 | 73.72 | **81.24** | 73.66 |
| 5 | 72.95 | **83.07** | 79.33 | 81.63 |
| 6 | 80.87 | 83.20 | 83.19 | **87.02** |
| 7 | 88.97 | **95.91** | 91.18 | 89.57 |
| 8 | 97.55 | 98.09 | 36.45 | **98.11** |
| 9 | 94.25 | 96.86 | **98.29** | 97.20 |
| 10 | 40.64 | 81.56 | **88.46** | 86.42 |
| 11 | 93.42 | 98.57 | **98.98** | 93.19 |
| 12 | 83.00 | 90.91 | **92.21** | 87.75 |
| 13 | 80.76 | 90.87 | **94.12** | 91.60 |
| 14 | 98.19 | **98.49** | 13.51 | 98.49 |
| 15 | 28.32 | 28.47 | **29.06** | 28.96 |
| 16 | 98.40 | **99.05** | 18.65 | 99.01 |
| 17 | 87.75 | **98.21** | 98.21 | 96.44 |
| 18 | 5.50 | **83.07** | 3.79 | 79.70 |
| 19 | 64.25 | **88.36** | 85.72 | 84.13 |
| Mean | 73.34 | **87.04** | 71.29 | 86.33 |
| Mean-Rank | 3.65 | **2.00** | 2.08 | 2.27 |
| Top-3 | 6 | **20** | 16 | 18 |
| Top-1 | 0 | 6 | **9** | 3 |

# G   Additional experiments

In this section, we present additional ablation studies and experiments related to ZEUS, including analyses regarding its robustness, architectural choices, clustering strategies and performance in special-case scenarios.

## G.1   Robustness to noise and outliers

In order to assess the robustness of our model under noise and outlier conditions, we consider two alternative evaluation setups:

- **ZEUS + noise**, where we add Gaussian noise with a standard deviation of 0.05 to each data point before the ZEUS forward pass,
- **ZEUS + anomalies**, where we introduce a set of global anomalies corresponding to $5\%$ of the dataset size. Each anomaly is generated by uniformly sampling numerical features from the range $(-1, 1)$ and randomly selecting categorical values. The resulting dataset, containing both ground truth data points and anomalies, is then passed through our model. Final clustering quality is measured only on the ground truth points, for which the true class assignments are known.

Table 21 displays the outcomes of these experiments, alongside the performance of the vanilla ZEUS model.

Table 21: Impact of Gaussian noise and anomalies on clustering performance.

| Metric | Dataset group | ZEUS+noise | ZEUS+anomalies | ZEUS |
|--------|---------------|------------|----------------|------|
| ARI | Real | 55.88 | 55.83 | **57.43** |
| ARI | Syn. Gauss. | 88.01 | 88.21 | **89.03** |
| ARI | Syn. Transf. | 84.82 | 85.17 | **86.33** |
| Rank | Real | 2.24 | 1.93 | **1.84** |
| Rank | Syn. Gauss. | 2.45 | 2.10 | **1.45** |
| Rank | Syn. Transf. | 2.45 | 2.30 | **1.25** |

Overall, the average performance in the perturbed setups is not significantly lower than that of the vanilla ZEUS model, highlighting a degree of stability of our method in the presence of noise and outliers.

## G.2   Impact of token dimension

The choice of token dimension also has a substantial influence on the performance and generalizability of ZEUS. This is confirmed by an experiment in which we evaluated dimensions of 32, 128, 512, and 768 while keeping the standard setup unchanged. Table 22 reports the clustering performance of these variants, denoted as *ZEUS-32*, *ZEUS-128*, *ZEUS-512*, and *ZEUS-768*, respectively.

Table 22: Assessment of different token dimensions in the ZEUS architecture.

| Metric | Dataset group | ZEUS-32 | ZEUS-128 | ZEUS-512 | ZEUS-768 |
|--------|---------------|---------|----------|----------|----------|
| ARI | Real | 44.10 | 47.81 | **57.43** | 50.79 |
| ARI | Syn. Gauss. | 69.50 | 83.04 | 89.03 | **91.92** |
| ARI | Syn. Transf. | 67.57 | 81.62 | 86.33 | **87.21** |
| Rank | Real | 2.90 | 2.96 | **1.90** | 2.25 |
| Rank | Syn. Gauss. | 4.00 | 3.00 | 1.78 | **1.23** |
| Rank | Syn. Transf. | 3.90 | 2.78 | 1.70 | **1.63** |

As one can infer, increasing the dimensionality leads to a consistent improvement in performance for synthetic data, whereas for OpenML datasets, there is a drop between dimensions 512 and 768, which may be caused by overfitting to the prior.

### G.3 Effect of the output clustering method

Since ZEUS is a method for learning representations suitable for clustering, in principle, any clustering algorithm could be used to separate the resulting clusters. However, our approach has the additional feature of being inherently designed to form spherical clusters; therefore, $K$-Means (*ZEUS-KM*) should, in theory, already be sufficient. The primary advantage of $K$-means lies in its speed, which is a crucial factor for us during inference. Moreover, as demonstrated in the Table 23, its clustering quality (measured by ARI) is comparable to the more computationally intensive Gaussian Mixture Model (*ZEUS-GMM*). For reference, we also include results from a simplified variant of GMM with fixed identity covariance matrices (*ZEUS-SGMM*), as employed in Section 3.3. All experiments are conducted using the same checkpoint and datasets as in Section 3.2.

Table 23: Evaluation of clustering methods applied to the learned ZEUS representations.

| Metric | Dataset group | ZEUS-KM | ZEUS-SGMM | ZEUS-GMM |
|--------|---------------|---------|-----------|----------|
| ARI | Real | **57.43** | 57.33 | 56.45 |
| ARI | Syn. Gauss. | 89.03 | 88.18 | **89.46** |
| ARI | Syn. Transf. | 86.33 | 85.57 | **87.22** |
| Rank | Real | **1.94** | 2.04 | 2.01 |
| Rank | Syn. Gauss. | 1.90 | 2.33 | **1.78** |
| Rank | Syn. Transf. | 2.13 | 2.35 | **1.53** |

Analysis of the average ARI scores indicates that ZEUS's output is well-suited to the evaluated approaches, as there are no significant differences in their results. Deep clustering techniques such as DEC or IDEC could also potentially be employed. Nevertheless, their reliance on intensive, dataset-specific training contradicts ZEUS's objective of delivering efficient, zero-shot representations.

### G.4 Integrating TabICL transformer into ZEUS

To mitigate one of the main limitations of our approach, namely the restricted number of input features and samples, we investigate whether it is feasible to replace the original TabPFN transformer architecture with a more recent design, such as the one used in TabICL [18]. Specifically, we adopt the TabICL transformer with its default hyperparameters and integrate it into our unsupervised pipeline (*ZEUS + TabICL*) in exactly the same manner as described in Section 3.1. The resulting clustering quality, alongside a comparison with the original ZEUS model, are presented in Table 24.

Table 24: Performance of ZEUS vs. its variant with the TabICL transformer backbone.

| Metric | Dataset group | ZEUS | ZEUS + TabICL |
|--------|---------------|------|---------------|
| ARI | Real | **57.43** | 52.62 |
| ARI | Syn. Gauss. | 89.03 | **89.81** |
| ARI | Syn. Transf. | **86.33** | 85.97 |
| Rank | Real | **1.38** | 1.62 |
| Rank | Syn. Gauss. | 1.60 | **1.40** |
| Rank | Syn. Transf. | 1.75 | **1.25** |

Overall, *ZEUS+TabICL* performs competitively with the original ZEUS across all evaluated data collections and even surpasses it in rank on synthetic data. Nonetheless, its average ARI on real-world datasets is noticeably lower. Future work should investigate whether this is due to random seed variability, our incomplete understanding of the TabICL architecture, potential overfitting to the prior, or the need for a more expressive prior.

### G.5 Special-case scenarios

Finally, we examine the behavior of our model in several extreme scenarios. To this end, we construct four synthetic two-dimensional datasets designed to highlight specific challenges:

- **Cluster-Imbalance**: Three clusters with 1000, 300, and 50 samples, respectively.

- **Covariance-Scaled**: Three equally sized clusters (500 samples each) with covariance matrices scaled by factors of 1, 6, and 36.
- **Moons**: Generated using *make_moons* from *scikit-learn* with 1000 samples and 0.1 noise.
- **Connection**: Two spherical Gaussians linked by a degenerate linear Gaussian.

We report a comparison of ZEUS performance against $K$-Means and GMM in Table 25.

Table 25: Comparison of clustering quality on synthetic special-case datasets, assessed by the ARI metric (higher scores indicate better quality).

|  | KM | GMM | ZEUS |
|---|---|---|---|
| Cluster-Imbalance | 50.12 | 96.75 | **96.80** |
| Covariance-scaled | 79.68 | **96.45** | 94.72 |
| Moons | 48.67 | 49.23 | **98.80** |
| Connection | 72.76 | **89.30** | 83.16 |

In summary, ZEUS outperforms $K$-means across all cases and performs on par with GMM. Notably, on the Moons dataset, ZEUS is the only method that successfully separates the clusters. This suggests that these synthetic examples do not represent genuine failure cases for our approach. A more thorough analysis of datasets where ZEUS underperforms remains an important direction for future work.

## H   Licensing and Third-Party Assets.

Our method builds upon the publicly available TabPFN codebase, which is released under the Apache 2.0 License. All real-world datasets used in the evaluation are sourced from OpenML.org, a platform hosting open datasets for machine learning research; all datasets used are publicly accessible and labeled as open data, although individual datasets may be subject to specific licenses (e.g., CC-BY or similar). We ensured that no proprietary or restricted datasets were used. Our released code is provided under the Apache 2.0 License and includes full instructions to reproduce the experiments.

**Broader Impact.**   Although ZEUS is designed as a general-purpose tool for clustering tabular data, its use in high-stakes domains like healthcare, finance, or criminal justice carries inherent risks. In such contexts, automated grouping of individuals - especially without labels or fairness constraints - can lead to biased or opaque outcomes. We therefore recommend that any deployment of ZEUS in sensitive applications be accompanied by fairness assessment and expert review.

