# OpenReview forum: "ZEUS: Zero-shot Embeddings for Unsupervised Separation of Tabular Data"
_NeurIPS.cc/2025/Conference — NeurIPS 2025 poster_

### Official Review · Reviewer_gk7v · 2025-06-19

**Clarity:** 3
**Significance:** 3
**Originality:** 4
**Rating:** 5
**Confidence:** 4

**Summary:**

The authors propose an unsupervised approach to learn embeddings for zero-shot clustering of the tabular data. The main ingredients of their approach are: 1) pre-train using synthetic data as in TabPFN 2) Transformer architecture for the backbone 3) loss based on GMM. In a series of experiments, the authors demonstrate the applicability of their approach to the

**Questions:**

## Understanding loss
1. From your final objective given in (3), I have doubts about the difficulty of the optimization and the importance of good initialization. Do you keep updating $\hat c_k$ at each iteration?
2. What role do $\hat \pi$'s play? In unsupervised settings, we use EM to fit GMM, so $\hat \pi$'s are updated each E-step. In your case, they seem fixed, defined by the training label.

## Architecture
1. There is little that is said about the architecture in the paper. The authors mention that they re-use TabPFN backbone in Section 3.1. However, for TabPFN the training label is passed together with the features. There's also cross-attention between training and test sets. Could the authors elaborate more on the specific architectural adjustments?
2. Major drawback of TabPFN is the context length, which is currently limited at around 10,000 samples for TabPFN V2. Recent works such as TabFlex or Mixture of ICP have explored addressing this concern via subsampling or compressing the data. Are these approaches portable to ZEUS?

**Ethical Concerns:**

["NO or VERY MINOR ethics concerns only"]

**Final Justification:**

The authors have effectively answered all my questions and concerns during the rebuttal.

**Limitations:**

yes

**Quality:**

3

**Strengths And Weaknesses:**

## Context
Deep learning for tabular (heterogeneous) data has seen large number of works, which can be roughly grouped into architecture improvements (TabNet, NODE), unsupervised learning (SwitchTab, Scarf), retrieval-augmented models (TabR, ModernNCA), and foundational models (TabGPT, TabPFN). The latter, TabPFN, is the latest work on the foundational models for tabular data that has demonstrated an almost magical ability to generalize to previously unseen datasets in a zero-shot fashion via in-context learning. This work builds on the idea of TabPFN and extends it to the tabular clustering problem.

## Strength
1. **Empirical results.** The method clearly outperforms other approaches based on shallow models (e.g. GMM) and autoencoders (e.g. AE-GMM, G-CEALS). The authors provide an ablation study to elucidate design choices of the loss. In addition, they evaluate model calibration, which is useful in many settings.
2. **Impact in practice.** The authors formulate the problem as a GMM. Therefore, the predicted embeddings are organized spherically in the latent space, which makes them directly amenable to application with downstream algorithms such as KNN or K-Means with minimal adaptation.
3. **Novelty.** To my knowledge, this is the first zero-shot embedding algorithm for tabular data.

## Weaknesses
1. **Incomplete comparison.** **Firstly**, the TabPFN repo has extensions, one of which is the unsupervised learning of embeddings: https://github.com/PriorLabs/tabpfn-extensions/tree/main/src/tabpfn_extensions/embedding. I'm not sure which work has appeared first, but nevertheless, I feel it would be a valuable comparison to understand the difference between TabPFN and ZEUS embeddings. **Secondly**, there are prior works on the unsupervised learning for tabular data such as Scarf or SwitchTab -- these are not zero-shot, but it would be nice to include some reference (or benchmark) to them to understand how they differ from the current work.
2. **Failure cases not explained.** If we take this work apart, the technical novelty lies in prior dataset generation and loss formulation. Both are formulated / generated using GMM. My complaint is that the section on the results may not accurately depict the real-world performance: 1) simulated / transformed datasets also follow GMM -- all the model needs to learn here is how to learn identity or inverse (of ResNet) transformation 2) real datasets are chosen to have ARI > 0.4 by at least one of the methods. Related to 2), there are multiple  problems: a) the threshold for cutoff appears a bit arbitrary b) all of the methods (AE, GMM) build on top of Gaussian assumption -> therefore selected real datasets are likely to have the Gaussian bias as well c) the problem in (b) could be addressed if the authors include other methods e.g. TabPFN or SwitchTab. In general, it would be great to see a stress test of the model to understand when it fails. Let's say there are no clusters, or if there is a severe class imbalance, or there is a mode connecting, or the geometry of clusters is very difficult.

---

> ### Author Rebuttal · Authors · 2025-07-31
>
> We thank the reviewer for their positive evaluation and for the insightful and constructive comments. We especially appreciate the suggestions regarding additional comparisons, architectural clarity, and stress-testing the method. In response, we have conducted new experiments and added clarifications, which we summarize below and will include in the final version of the paper.
>
> > 1. Incomplete comparison. Firstly, the TabPFN repo has extensions, [..] Secondly, there are prior works on the unsupervised learning for tabular data such as Scarf or SwitchTab -- these are not zero-shot, but it would be nice to include some reference (or benchmark) to them to understand how they differ from the current work.
>
> We appreciate the suggestion to compare ZEUS with unsupervised representation learning approaches in the context of clustering task. Accordingly, we incorporate both TabPFN-Unsupervised and SCARF into our evaluation.
>
> For TabPFN-Unsupervised, while the linked method relies on label information from the training set to generate representations, the same repository also includes an `unsupervised` folder containing the `TabPFNUnsupervisedModel` class. We use its `get_embeddings_per_column` method to extract per-column representations, which we then average across columns to obtain a final row-level representation. Unfortunately, we were unable to use the `get_embeddings` function due to the following error:
>
> NotImplementedError(
> "This method is not implemented currently. During the main TabPFN refactor this functionality was removed, [...]"
> )
>
> In the case of SCARF, we follow the `example.ipynb` Jupyter notebook provided in the original repository. For both SCARF and TabPFN-Unsupervised, we cluster their learned representations using k-means, analogously to our approach in ZEUS. Additionally, we included IDC in our evaluation, as suggested by **Reviewer 2ngo**.
>
> |Metric| Dataset group |IDC|TabPFN-Uns.|SCARF|GMM|KM|AE-KM|AE-GMM|DEC|IDEC|G-CEALS | ZEUS|
> |-|-|-|-|-|-|-|-|-|-|-|-|-|
> |ARI|Real|52.28|31.32|26.95|48.49|55.54|51.43|53.56|55.93|54.57|40.37|57.43|
> |ARI|Syn. Gauss.|66.43|55.97|8.32|76.93|89.90|81.26|81.40|89.36|82.57|62.84|89.03|
> |ARI|Syn. Transf.|66.78|15.66|2.48|75.88 |75.04 |60.45|71.29|79.94|61.26|49.17|86.33|
> |Rank|Real|5.62|8.22|8.85|5.65|4.69|5.72|5.24|4.69|5.01|8.18|4.13|
> |Rank|Syn. Gauss.|7.95|8.75|11.00|3.65|2.65|5.65|5.60|3.20|5.70|8.90|2.95|
> |Rank|Syn. Transf.|6.05|9.75|11.00|3.50|4.80|6.85|4.50|3.20|6.35|7.80|2.20|
>
> Based on the results in the table, one can infer that methods like SCARF or TabPFNUnsupervised are more suited for feature extraction, which can then be fine-tuned for classification tasks. Without additional regularization components during training, or specialized fine-tuning such as with DEC, we are unable to effectively separate these representations.
>
> > 2. Failure cases not explained. If we take this work apart, the technical novelty lies in prior dataset generation and loss formulation. Both are formulated / generated using GMM. My complaint is that the section on the results may not accurately depict the real-world performance: [...]. In general, it would be great to see a stress test of the model to understand when it fails. Let's say there are no clusters, or if there is a severe class imbalance, or there is a mode connecting, or the geometry of clusters is very difficult.
>
> Firstly, we acknowledge that benchmarking clustering performance in realistic failure scenarios is inherently difficult, particularly for small and medium-sized tabular datasets. To our knowledge, there is no widely accepted benchmark that provides well-defined clustering ground truth beyond class labels, which may not reflect true cluster structure.
>
> Secondly, we are grateful to the reviewer for the insightful suggestion to examine the failure cases of our model. While such analysis requires careful consideration and time, we have taken initial steps in that direction. Specifically, we constructed four synthetic 2D scenarios:
> - **Cluster-Imbalance**: Three clusters with 1000, 300, and 50 samples, respectively.
> - **Covariance-Scaled**: Three equally sized clusters (500 samples each) with covariance matrices scaled by factors of 1, 6, and 36.
> - **Moons**: Generated using make_moons from scikit-learn with 1000 samples and 0.1 noise.
> - **Connection**: Two spherical Gaussians linked by a degenerate linear Gaussian.
>
> | | Cluster-Imbalance | Covariance-scaled | Moons  | Connection   |
> |-|-|-|-|-|
> | KM   | 0.5012| 79.68| 48.67 | 72.76 |
> | GMM  | 0.9675| 96.45| 49.23 | 89.30 |
> | ZEUS | 0.9680| 94.72| 98.80 | 83.16 |
>
> ZEUS outperforms K-means across all cases and performs on par with GMM. Notably, on the Moons dataset, ZEUS is the only method that successfully separates the clusters. This suggests that these synthetic examples do not represent genuine failure cases for our approach. A more thorough analysis of datasets where ZEUS underperforms remains an important direction for future work.
>
> > Understanding loss
> > 1. From your final objective given in (3), I have doubts about the difficulty of the optimization and the importance of good initialization. Do you keep updating  $\hat{c}_k$ at each iteration?
>
> First of all, $\hat{c}_k$​ is estimated as the average representation of the points assigned to cluster $k$. Therefore, it is indeed updated dynamically and differs across datasets.
>
> As for initialization, it certainly plays a role in optimization process. A good initialization under a given random seed can lead to faster convergence and potentially better performance. Conversely, extremely poor initialization may result in instability or even complete collapse of pre-training. However,  we did not observe this to be a significant issue during our experiments. Nonetheless, investigating the impact of initialization strategies could be a potential direction for future work.
>
> > 2. What role do $\hat{\pi}$'s play? In unsupervised settings, we use EM to fit GMM, so $\hat{\pi}$'s are updated each E-step. In your case, they seem fixed, defined by the training label.
>
> Yes, the values of $\hat{\pi}$ are defined based on the training labels and are not updated in any other way. Nonetheless, introducing $\hat{\pi}$ helped slightly improve performance, particularly on real-world datasets. This improvement may stem from the fact that incorporating information about cluster balance increases the flexibility of the learning process and provides a beneficial learning bias. The comparison between the version with a prior (ZEUS) and the one without it (ZEUS-without-prior) in the loss function is as follows:
>
> | Metric | Dataset group | ZEUS   | ZEUS-without-prior |
> |-|-|-|-|
> |ARI|Real | 57.43 | 50.12|
> |ARI|Syn. Gauss.|89.03|88.42|
> |ARI| Syn. Transf.|86.33|83.35|
> | Rank|Real| 1.3088 | 1.6912|
> | Rank|Syn. Gauss. | 1.4000 | 1.6000|
> | Rank|Syn. Transf. | 1.2250 | 1.7750|
>
> > Architecture
> > 1. There is little that is said about the architecture in the paper. The authors mention that they re-use TabPFN backbone in Section 3.1. However, for TabPFN the training label is passed together with the features. There's also cross-attention between training and test sets. Could the authors elaborate more on the specific architectural adjustments?
>
> Similarly to TabPFN, we employ a linear embedding layer to convert each row of the table into a 512-dimensional token. However, since we are addressing an unsupervised problem, no training label embeddings are added to these tokens. Moreover, our setting does not involve a query set, and thus we apply only self-attention within the support set. The transformer architecture matches that of TabPFN: it consists of 12 attention blocks, each with 6 attention heads and GeLU as the activation function. At its output, we obtain a representation for each data point, which is then clustered using k-means. Notably, unlike TabPFN, no additional MLP decoder is applied after the transformer.
>
> > 2. Major drawback of TabPFN is the context length, which is currently limited at around 10,000 samples for TabPFN V2. Recent works such as TabFlex or Mixture of ICP have explored addressing this concern via subsampling or compressing the data. Are these approaches portable to ZEUS?
>
> Yes, we believe that architectures of TabPFN’s successors are transferable to ZEUS. To preliminarily test this, we explore the possibility of adapting the TabICL [1] architecture pipeline to our unsupervised clustering setting, following the same methodology as in the case of TabPFN (see response above). The resulting performance is shown below:
>
> | Metric | Dataset group | ZEUS   | ZEUS + TabICL |
> |-|-|-|-|
> |ARI|Real| 57.43 | 52.62|
> |ARI|Syn. Gauss.|89.03 | 89.81|
> |ARI|Syn. Transf.|86.33 | 85.97|
> |Rank|Real|1.3824 | 1.6176|
> |Rank|Syn. Gauss.|1.6000 | 1.4000|
> |Rank |Syn. Transf. |1.7500 | 1.2500|
>
> ZEUS + TabICL performs competitively with the original ZEUS across all evaluated data collections and even surpasses it in rank on synthetic data. Nonetheless, its average ARI on real-world datasets is noticeably lower. Future work should investigate whether this is due to random seed variability, our incomplete understanding of the TabICL architecture, potential overfitting to the prior, or the need for a more expressive prior.
>
> We are grateful for the reviewer’s thoughtful suggestions, which have led to a stronger and more complete version of the work. We hope the added comparisons (TabPFN-Unsupervised, SCARF), architectural clarifications, stress-test scenarios, and ablations help reinforce the contribution and reliability of ZEUS. We appreciate the reviewer’s positive assessment.
>
> ---
>
> [1] Qu, J., Holzmüller, D., Varoquaux, G., & Le Morvan, M. (2025). TabICL: A Tabular Foundation Model for In-Context Learning on Large Data. arxiv preprint

---

> > ### Comment · Reviewer_gk7v · 2025-08-03
> > **Response to Rebuttal**
> >
> > The authors have answered all of my questions and addressed the concerns. Overall, I believe this is a strong work and deserves to be included in the main conference.

---

### Official Review · Reviewer_3Vjr · 2025-06-27

**Clarity:** 3
**Significance:** 2
**Originality:** 2
**Rating:** 4
**Confidence:** 5

**Summary:**

The paper presents a transformer-based approach (called ZEUS) where a transformer is trained on datasets with ground truth cluster labels and learns to assign cluster labels to new datasets. Zeus has been compared against k-means, GMMs and deep clustering methods on several families of real and synthetic datasets.

**Questions:**

Q1. It is not clear how the real datasets have been separated into training and testing.
Q2. In several cases, ZEUS performance is very low (much lower that k-means and GMMs). This reduces confidence in the method’s validity.
Q3. It would be interesting to compare performance on datasets with discrete features only. Discrete features cause difficulties in k-means and GMMs.
Q4. For many real datasets the number of clusters is 2, which is quite small.
Q5. It should be emphasized that k-means is applied on the transformer output in order to obtain the final cluster labels.
Q6. Some results on the influence of noise and outliers would add value to the experimental study.

**Ethical Concerns:**

["NO or VERY MINOR ethics concerns only"]

**Final Justification:**

As emphasized in my discussion with the authors, I believe that the proposed method is interesting, however,  several open issues remain that need to be addressed. I have increased my score putting more emphasis on the novel aspects of the method.

**Limitations:**

The major limitations of the method are discussed in the paper: dependence on the synthetic datasets used for training and restrictions on the scale of the dataset.

**Paper Formatting Concerns:**

No concern.

**Quality:**

3

**Strengths And Weaknesses:**

Strengths
S1. ZEUS implements a novel idea on how to train a transformer using annotated clustering datasets in order to assign cluster labels on new datasets.
S2. The approach can accommodate mixed datasets with both real and categorical features.

Weaknesses
W1. Since supervised training is used, clustering performance strongly depends on the annotated datasets used for clustering.
W2. There are limits imposed by the input dimensionality of the transformer (dataset size, number of features)

---

> ### Author Rebuttal · Authors · 2025-07-31
>
> Reviewer 3Vjr (3, conf =5)
> We thank the reviewer for the detailed and insightful comments. We appreciate the recognition of the potential impact of ZEUS, and agree that its claims must be supported by strong comparisons, ablations, and a deeper analysis of edge cases. Below, we respond point-by-point and provide several additional experiments (e.g., discrete-only data, impact of synthetic priors, sensitivity to noise), which will be included in the final version of the paper.
>
> > W1. Since supervised training is used, clustering performance strongly depends on the annotated datasets used for clustering.
>
> We agree that the choice of training distributions has a strong influence on the resulting clustering behavior, and that this creates opportunities for further improvement. In our current work, we approach the problem from a generative perspective rather than by committing to a fixed clustering loss. Specifically, we adopt the widely used latent variable model (LVM) paradigm, where a discrete latent variable determines cluster membership. To make this framework applicable beyond simple Gaussian mixtures, we apply random neural transformations to the observed space, enabling a rich diversity of data distributions while preserving the underlying cluster structure.
> ZEUS is trained to reverse this generative process, allowing it to generalize across datasets without relying on predefined objectives like MSE. That said, we believe that incorporating more structured or domain-specific generative processes, or learning such transformations from real data, is a promising direction for future work.
>
> > W2. There are limits imposed by the input dimensionality of the transformer (dataset size, number of features)
>
> These limitations are primarily inherent to the architecture of the vanilla TabPFN model. However, recent research, including TabPFN2 [1], TabICL [2], TabFlex [3] has shown that such restrictions can be gradually mitigated. We have conducted preliminary tests indicating that the TabICL architecture can be possibly integrated into our pre-training pipeline. The corresponding results are presented in the table below:
>
>
> | Metric | Dataset group | ZEUS   | ZEUS + TabICL |
> |--------|---------------|--------|---------------|
> | ARI    | Real | 57.43 | 52.62        |
> | ARI    | Syn. Gauss.   | 89.03 | 89.81        |
> | ARI    | Syn. Transf.  | 86.33 | 85.97        |
> | Rank   | Real          | 1.3824 | 1.6176        |
> | Rank   | Syn. Gauss.   | 1.6000 | 1.4000        |
> | Rank   | Syn. Transf.  | 1.7500 | 1.2500        |
>
> ZEUS + TabICL performs competitively with the original ZEUS across all evaluated dataset groups and even surpasses it in average rank on synthetic data. However, its ARI on real-world datasets is slightly lower. This may stem from seed sensitivity, architectural factors, or limitations in how the prior interacts with more heterogeneous tabular distributions. Investigating these effects and improving compatibility with TabICL-style architectures is a promising avenue for future work.
>
> > Q1. It is not clear how the real datasets have been separated into training and testing.
>
>
> We clarify that real datasets are used only for testing. The model is trained exclusively on synthetically generated data. No samples from the real datasets are seen during training, and there is no fine-tuning. This setup ensures a strict zero-shot evaluation.
>
>
> > Q2. In several cases, ZEUS performance is very low (much lower that k-means and GMMs). This reduces confidence in the method’s validity.
>
>
> It is true that ZEUS underperforms on some individual datasets. However, no clustering method performs well across all settings, which is evident in the performance of other deep models that often fall behind k-means or GMMs on average. What sets ZEUS apart is its ability to generalize in settings where others fail.
> In particular, on synthetic datasets with transformed latent structures (Appendix Table 10), ZEUS outperforms all baselines by a large margin (~10 pp), showing its strength in realistic, non-Gaussian clustering scenarios with well-defined ground truth.
> For real-world datasets, we follow standard benchmarks where class labels are used as cluster references. While common in the literature, this is not always meaningful in practice, as classes do not necessarily align with cluster structure. Still, ZEUS ranks first on average and achieves ARI above 0.9 on datasets where many methods fail completely (ARI = 0) - for example, OpenML dataset 1462, as shown in Appendix Table 8. This demonstrates that ZEUS fills an important gap by succeeding where others collapse.
>
> > Q3. It would be interesting to compare performance on datasets with discrete features only. Discrete features cause difficulties in k-means and GMMs.
>
> The evaluated datasets (GuptaA_2019_PA and QinN_2014_PA) are presence/absence (binary) metagenomic profiles obtained from the curatedMetagenomicData [4] collection. Each feature indicates whether a specific microbial taxon is present in a given sample. The datasets are named after the first author and year of the corresponding publication, following the convention used in the original repository. Their high dimensionality, sparsity, and purely discrete nature make them particularly challenging for clustering methods based on continuous distances, such as k-means or GMMs.
> |                | KM    | ZEUS  |
> |----------------|-------|-------|
> | GuptaA_2019_PA | 12.55 | 31.16 |
> | QinN_2014_PA   | 35.14 | 41.43 |
>
> The table highlights the potential of ZEUS, which could likely be further enhanced by pre-training specifically on discrete features or by replacing one-hot encoding with alternative embedding methods.
>
> > Q4. For many real datasets the number of clusters is 2, which is quite small.
>
>
> It is true that many real-world datasets in our benchmark have only 2 classes, as they originate from standard binary classification tasks. However, approximately half of the datasets contain more than two classes, and we observe that ZEUS often performs especially well in these more complex multi-cluster scenarios. For example, on datasets such as 14, 16, 22, 35, 458, 1499, 4153, and 42585, ZEUS outperforms all other baselines, which highlights its strength in capturing more nuanced latent structures. Synthetic datasets, in particular, are predominantly multi-class and further demonstrate this advantage.
>
> > Q5. It should be emphasized that k-means is applied on the transformer output in order to obtain the final cluster labels.
>
> Thank you for pointing this out. We agree that this detail should be made explicit, and we will make sure to emphasize it clearly in the final version of the paper.
>
>
> > Q6. Some results on the influence of noise and outliers would add value to the experimental study.
>
>
> That is a great remark, and we thank the reviewer for highlighting it. We were able to conduct some initial experiment on this matter. In addition to the standard evaluation method (referred to as ZEUS), we introduce two additional variants:
> ZEUS + noise: Gaussian noise with a standard deviation of 0.05 is added to each data point before the ZEUS forward pass.
> ZEUS + anomalies: For each dataset, we introduce a set of global anomalies corresponding to 5% of the dataset size. These anomalies are constructed by uniformly sampling numerical features from the range (−1,1) and randomly selecting categorical values.
> The results are as follows:
>
> | Metric | Dataset group | ZEUS   | ZEUS+noise | ZEUS+anomalies |
> |--------|---------------|--------|------------|----------------|
> | ARI    | Real          | 57.43 | 55.88     | 55.83         |
> | ARI    | Syn. Gauss.   | 89.03 | 88.01     | 88.21         |
> | ARI    | Syn. Transf.  | 86.33 | 84.82     | 85.17         |
> | Rank   | Real          | 1.8382 | 2.2353     | 1.9265         |
> | Rank   | Syn. Gauss.   | 1.4500 | 2.4500     | 2.1000         |
> | Rank   | Syn. Transf.  | 1.2500 | 2.4500     | 2.3000         |
>
> As we can see, the average performance in the perturbed variants is not significantly lower than that of the vanilla ZEUS model, highlighting a degree of stability in the method.
>
> These additional experiments and clarifications provide a broader view of ZEUS’s strengths and limitations. We appreciate the reviewer’s thoughtful and technically grounded suggestions, which have helped improve the depth and rigor of our evaluation. We hope the extended results and stress tests address the reviewer’s concerns and support a more favorable assessment.
>
> ---
>
> [1] Hollmann, N., Müller, S., Purucker, L., Krishnakumar, A., Körfer, M., Hoo, S. B., Schirrmeister, R. T., & Hutter, F. (2025). Accurate predictions on small data with a tabular foundation model. Nature.
>
> [2] Qu, J., Holzmüller, D., Varoquaux, G., & Le Morvan, M. (2025). TabICL: A Tabular Foundation Model for In-Context Learning on Large Data. arxiv preprint
>
> [3] Zeng, Y., Dinh, T., Kang, W., & Mueller, A. C. (2025). TabFlex: Scaling Tabular Learning to Millions with Linear Attention. arxiv preprint
>
> [4] Pasolli E, Schiffer L, Manghi P, Renson A, Obenchain V, Truong D, Beghini F, Malik F, Ramos M, Dowd J, Huttenhower C, Morgan M, Segata N, Waldron L (2017). “Accessible, curated metagenomic data through ExperimentHub.” Nat. Methods, 14(11), 1023–1024. ISSN 1548-7091, 1548-7105,

---

> ### Comment · Reviewer_3Vjr · 2025-08-04
> **reply to author rebuttal**
>
> I thank the authors for clarifications and additional experimental results that strengthen the manuscript. However, I believe that weakness W1 (generalizing from synthetic annotated datasets used for training to real datasets), which is actually the main idea introduced in this work, is not adequately addressed, although there are some promising initial experimental results. The dependence on adhoc decisions - such as the datasets selected for training and the random neural transformations used - reduce the confidence on the quality of the clustering solution.  Perhaps some theoretical results would make this research direction more promising.

---

> > ### Author Response · Authors · 2025-08-05
> > **Theoretical clarification**
> >
> > According to the reviewer's remark, we would like to clarify our reasoning from theoretical perspective. **Our theoretical result (see below) show that our procedure of generating synthetic ground-truth datasets for clustering generalizes to real datasets, which come from a mixture of arbitrary distributions**.
> >
> > The mixture model is a typical way to represent datasets with natural clustering structures. We would like to emphasize that in the unsupervised setting (like clustering task), we always need to make prior assumptions on the clustering structure, we are looking for (either by a loss function or using a probability model of clusters). Our procedure was designed so that to cover clusters, which represent the mixture of arbitrary distributions (not only Gaussians). We will include this theoretical analysis to Section 2.2.
> >
> > **Theorem (Univeral Approximation Theorem for Mixture Distributions)**
> > Let $P = \sum_{i=1}^k \pi_i P_i$ be a mixture of distributions on $\mathbb{R}^d$, where each $P_i$ is a probability distribution, and $\pi_i > 0$, $\sum_{i=1}^k \pi_i = 1$.
> >
> > Then there exists a mixture of Gaussians $Q = \sum_{i=1}^k \pi_i \mathcal{N}(\mu_i, \Sigma_i)$ and a neural network $F$ such that we can approximate $P$ with arbitrarily small error by the pushforward measure $F_\\# Q$. Additionally, we can select $F$ so that $F_\\# \mathcal{N}(\mu_i,\Sigma_i)$ approximates $P_i$ with arbitrarily small error.
> >
> > **Proof** We proceed in several steps:
> >
> > *Step 1: Approximation of individual components.*
> >
> > Making use of the universal approximation theorem, for each component $P_i$ there exists a neural network $F_i$ which transforms an arbitrary Gaussian $Q_i = \mathcal{N}(\mu_i,\Sigma_i)$ into $P_i$, i.e. $F_{i \\#} Q_i \approx P_i$.
> >
> > *Step 2: Constructing a unified neural network.*
> >
> > We now define a single neural network $F$ that can represent all the $F_i$ networks. Define $F : \mathbb{R}^d \times \{1, \dots, k\} \to \mathbb{R}^d$ such that $F(z, i) \approx F_i(z)$. This can be implemented by encoding the index $i$ as a one-hot vector or embedding and concatenating it to $z$, enabling $F$ to learn the behavior of each $F_i$ conditioned on the index $i$. The network $F$ can be extended to the domain $\mathbb{R}^d \times \mathbb{R}$ in an arbitrary way.
> >
> > *Step 3: Approximation of the full mixture.*
> >
> > Since each $F_{\\#} \mathcal{N}(\mu_i, \Sigma_i)$ approximates $P_i$, the mixture $F_{\\#} Q = \sum_{i=1}^k \pi_i F_{\\#} \mathcal{N}(\mu_i, \Sigma_i)$ approximates $P = \sum_{i=1}^k \pi_i P_i$.
> >
> > Thus, we have constructed a mixture of Gaussians $Q$ and a neural network $F$ such that $F_{\\#} Q$ approximates $P$, and each component $F_{\\#} \mathcal{N}(\mu_i, \Sigma_i)$ approximates $P_i$ arbitrarily well. This completes the proof.
> >
> > **Conclusion**
> >
> > We are grateful to the reviewer for suggesting a theoretical analysis. This addition will undoubtedly strengthen our contribution.
> >
> > **Remark**
> >
> > For transparency, we keep our theoretical formulation less formal. We can also use more precise notion of approximation if needed.

---

> > > ### Comment · Reviewer_3Vjr · 2025-08-06
> > >
> > > I thank the authors for the additional theoretical result, however it does not resolve my 'generalization' concern. Typically, clustering methods rely on models (or assumptions about 'cluster structure') and operate  towards optimizing appropriate criteria or loss functions. Therefore a solution is 'optimal' with respect to the specified objective. The proposed approach does not seem to follow this direction. It is not clear to me what are the cluster structures that the method is able to discover and, more importantly, there is no optimality criterion that drives the discovery of such clustering solutions.
> > >
> > > This is the reason that I consider the method as 'adhoc', although providing good results in many datasets (however in some cases it is inferior to kmeans). The method relies on the generalization ability of a model trained in a supervised way. However, generalization depends on the datasets used for training which is a highly adhoc decision.
> > >
> > > In general, the paper presents a novel idea, but requires significant additional research to become a convincing widely accepted clustering methodology. Nevertheless, focusing on novelty and positive opinion of other reviewers, I will increase my score.

---

> > > > ### Author Response · Authors · 2025-08-06
> > > >
> > > > We are grateful to the Reviewer for initiating an inspiring discussion. We would like to follow up with a clarification of our reasoning in a less formal way.
> > > >
> > > > We agree that typical clustering models focus on finding clusters which minimize a given objective. As the Reviewer mentioned, optimal solutions minimize these objectives. However, this does not necessarily mean that the clusters found in this way will be 'optimal' or 'useful' to the user. There are numerous objectives in the literature, and there is no consensus on which one consistently allows us to discover the true clustering structure in data. This motivates us to follow a completely different direction—one that is not based on any specific objective function, as such objectives are inherently subjective and cannot offer a universal solution.
> > > >
> > > > In our work, we asked how to describe general clustering structures without tying them to a specific objective. Our idea is that most clustering structures can be described by mixtures of arbitrary probability distributions. We do not restrict ourselves to any particular family of distributions (e.g., Gaussians), but instead allow for highly flexible ones. Current advances in generative modeling (e.g., diffusion models, GANs, VAEs) demonstrate that practically any dataset can be described by a neural network transformation of a Gaussian distribution—this forms the foundation of our theoretical result. Following this reasoning, our procedure for generating synthetic training datasets can represent arbitrary clustering structures based on mixture models. While we do impose some restrictions on the neural network transformation, it still covers a significantly broader class of mixture models than previous approaches.
> > > >
> > > > In our model, we do not define a loss function for detecting individual components from these arbitrary mixtures. Instead, we work in the opposite direction: we generate a wide spectrum of mixtures with known ground-truth labels and formulate a supervised pre-training task. If the model is trained correctly, it will be able to detect similar clustering structures as those shown during pre-training. For instance, in Table 1, ZEUS detects Gaussian clusters better than GMM (which is designed specifically for this task), and detects transformed Gaussians significantly better than any other method. This demonstrates that our model was trained correctly and is able to detect structures similar to those seen during pre-training.
> > > >
> > > > Clearly, its performance on real-world data may not appear as strong at first glance. However, those are classification datasets, and their classes often do not correspond to meaningful clusters (i.e., subsets separated by low-density regions or lying on lower-dimensional manifolds). Without access to ground-truth labels, such classes are often undetectable. This is why we place greater emphasis on results from data representing transformed Gaussians.
> > > >
> > > > We hope this explanation offers a clearer view of our model. Once again, thank you for the in-depth reading of our paper and for reconsidering your decision.

---

### Official Review · Reviewer_2ngo · 2025-07-01

**Clarity:** 4
**Significance:** 2
**Originality:** 3
**Rating:** 5
**Confidence:** 2

**Summary:**

The paper proposes a framework for unsupervised representation learning for tabular data, such that the resulting representations can be used for clustering datasets in a zero-shot manner. To achieve this, the proposed method (ZEUS) leverages a novel method for generating synthetic data for pre-training, and a set of losses that promote the desirable properties in the learned embedding space. The resulting model leads to strong clustering results in the evaluations conducted.

**Questions:**

My pripary concern is, as stated in the Weaknesses, related to the lack of experiments and comparisons in the paper. I would urge the authors to include more compared methods and ablations in the paper to strengthen the paper's findings (e.g. if more methods were included it might highlight the sifnificance of ZEUS' results). Alternatively, I would urge the authors to justify why works such as the ones I mentioned earlier were not included.

**Ethical Concerns:**

["NO or VERY MINOR ethics concerns only"]

**Final Justification:**

The authors have responded to the issues I raised in my review. Μy main concerns remain the architectural differences with competing methods and the (in my opinion) not entirelly convincing gains, but I appreciate the novel approach and the contribution regarding the training data mix and their impact. I therefore raise my recommendation to accept.

**Limitations:**

Yes

**Paper Formatting Concerns:**

No formatting issues.

**Quality:**

3

**Strengths And Weaknesses:**

Strengths:
1. The proposed methodology is novel, both regarding the synthetic sample generation method and the proposed losses.
2. ZEUS indeed outperforms alternative methods in terms of the quality of the generated clusters.
3. The paper is well written and its methodology and motivation is clear.

Weaknesses:
1. The results presented in Tab. 1 & 2 and Appendix Tab. 4-7 show that, on the aggregate, ZEUS does better than the compared methods. However, it is by small margins that undermine the impact of the proposed method.
2. The evaluation comparisons are lacking. Most methods compared are naive (e.g. K-Means), and more advanced methods (e.g. G-CEALS) may be at a disadvantage given the size of the models involved: my understanding is that the autoencoder used by G-CEALS is much smaller and less powerfull than the transformer architecture used by ZEUS, which means one cannot be sure how much of ZEUS' gains are related to the architecture as opposed to the method. Finally, it is unclear to my why methods such as DEPICT and IDC are not included in the comparisons.
3. The ablations should be more thorough. Most importantly, I found no study of the impact of using different synthetic data (Gaussian/Categorical/NN transformed) in the training mix, despite that being a key contribution/component of the paper.

---

> ### Author Rebuttal · Authors · 2025-07-31
>
> We thank the reviewer for the thoughtful and detailed feedback. We appreciate the recognition of ZEUS’s core contribution and agree that stronger experimental comparisons and more thorough ablations are essential to supporting its claims. In response, we have expanded our evaluation with additional experiments, which we summarize below and will include in the final version of the paper.
>
> > 1. The results presented in Tab. 1 & 2 and Appendix Tab. 4-7 show that, on the aggregate, ZEUS does better than the compared methods. However, it is by small margins that undermine the impact of the proposed method.
>
> We respectfully disagree with the claim that the gains reported in Tables 1 and 2 are marginal and insufficient to support the impact of our method. Clustering is inherently an unsupervised task, and without a clearly defined target it is difficult to determine which clustering is "better." Many real-world datasets commonly used in the literature were originally constructed for classification, and their class labels often do not correspond to well-separated clusters. As a result, most methods struggle to recover them. Despite this, ZEUS achieves the highest ARI on real-world data (57.43) and the best average rank (3.60), outperforming all compared baselines.
>
> To provide a more interpretable evaluation, we also include synthetic datasets with well-defined ground-truth clusters, based on latent variable models and mixture components. In these more structured scenarios, ZEUS shows clear and consistent advantages. We also include Gaussian mixtures as a sanity check — these are deliberately simple and yield similar results across methods, which naturally reduces visible margins.
>
> Overall, ZEUS demonstrates strong and reliable performance across diverse clustering regimes, including both ill-posed real-world data and well-defined synthetic tasks.
>
> > 2. The evaluation comparisons are lacking. Most methods compared are naive (e.g. K-Means), and more advanced methods (e.g. G-CEALS) may be at a disadvantage given the size of the models involved: my understanding is that the autoencoder used by G-CEALS is much smaller and less powerfull than the transformer architecture used by ZEUS, which means one cannot be sure how much of ZEUS' gains are related to the architecture as opposed to the method.
>
> For all deep clustering baselines, we used the implementations provided by the authors and followed their recommended hyperparameter selection strategies. For instance, the embedding dimension was chosen from the set {5, 10, 15, 20}, as suggested in the GCEALs paper, and we adhered to this choice, as it was deemed beneficial for those baselines. ZEUS contains 25.78 million parameters because of its large-scale pretraining across millions of datasets in a zero-shot regime. Baseline methods are trained on individual datasets but still use autoencoders with over 2.5 million parameters.
>
> As shown in the table below, increasing the size of the autoencoder from hidden layers [500,500,2000] (GCEALS-2.5M) to [500,500,2000,2000,2000] (GCEALS-10.5M) actually leads to worse performance on both real-world and synthetic Gaussian datasets. This suggests that simply scaling up the model can lead to some form of overfitting. Furthermore, if increasing the autoencoder’s size alone were beneficial, the original authors would have likely reported this. Nonetheless, exploring Transformer-based variants of DEC or GCEALs might be worthwhile, though such extensions go beyond the scope of this study.
>
> | Metric | Dataset group | GCEALS-2.5M | GCEALS-10.5M |
> |--------|---------------|-----------|------------|
> | ARI    | Real          | 40.37    | 26.91     |
> | ARI    | Syn. Gauss.   |62.84    | 56.50     |
> | ARI    | Syn. Transf.  | 49.17    | 50.67     |
> | Rank   | Real          | 1.2059    | 1.7941     |
> | Rank   | Syn. Gauss.   | 1.2000    | 1.8000     |
> | Rank   | Syn. Transf.  | 1.5000    | 1.5000     |
>
> In the case of GCEALS, the primary issue lies in its instability across different random seeds. The method exhibits high variance, which negatively affects the average performance. For illustration, the standard deviation of ARI scores on selected datasets is as follows:
>
> | Dataset group | ID    | GCEALS-STD |
> |---------------|-------|------------|
> | Real          | 15    | 9.65       |
> | Real          | 458   | 10.45      |
> | Real          | 187   | 30.82      |
> | Real          | 42261 | 23.64     |
> | Syn. Gauss.   | 3     | 10.74      |
> | Syn. Gauss.   | 8     | 16.99      |
> | Syn. Gauss.   | 10    | 44.1       |
> | Syn. Gauss.   | 17    | 7.01       |
> | Syn. Transf.  | 2     | 15.04      |
> | Syn. Transf.  | 7     | 11.65      |
> | Syn. Transf.  | 14    | 8.33       |
> | Syn. Transf.  | 19    | 25.75      |
>
> In general, finding the optimal configuration for dozens of hyperparameters in an unsupervised setting is inherently challenging due to the absence of labels. Our method offers a clear advantage in this regard, as its zero-shot nature allows it to avoid such issues during inference.
>
> > Finally, it is unclear to me why methods such as DEPICT and IDC are not included in the comparisons.
>
> Indeed, you are right, that IDC should have been included in our evaluation. According to the tables in the GCEALS paper, DEPICT is among the weaker methods and is also 4 to 5 times slower than GCEALS or DEC. Therefore, unfortunately, due to limited time and computational resources, we decided to exclude it from our experiments. However, in addition to our baseline methods, we also include TabPFN-Unsupervised and SCARF, as suggested by Reviewer gk7v. Similar to ZEUS, we cluster their learned representations using k-means. The clustering quality results are presented below:
>
> |Metric| Dataset group |IDC|TabPFN-Uns.|SCARF|GMM|KM|AE-KM|AE-GMM|DEC|IDEC|G-CEALS | ZEUS|
> |-|-|-|-|-|-|-|-|-|-|-|-|-|
> |ARI|Real|52.28|31.32|26.95|48.49|55.54|51.43|53.56|55.93|54.57|40.37|57.43|
> |ARI|Syn. Gauss.|66.43|55.97|8.32|76.93|89.90|81.26|81.40|89.36|82.57|62.84|89.03|
> |ARI|Syn. Transf.|66.78|15.66|2.48|75.88 |75.04 |60.45|71.29|79.94|61.26|49.17|86.33|
> |Rank|Real|5.62|8.22|8.85|5.65|4.69|5.72|5.24|4.69|5.01|8.18|4.13|
> |Rank|Syn. Gauss.|7.95|8.75|11.00|3.65|2.65|5.65|5.60|3.20|5.70|8.90|2.95|
> |Rank|Syn. Transf.|6.05|9.75|11.00|3.50|4.80|6.85|4.50|3.20|6.35|7.80|2.20|
>
> For IDC, we adhere to the setup and guidelines outlined in the official repository's Jupyter notebooks.
>
> > 3. The ablations should be more thorough. Most importantly, I found no study of the impact of using different synthetic data (Gaussian/Categorical/NN transformed) in the training mix, despite that being a key contribution/component of the paper.
>
> Indeed, such an analysis would significantly strengthen our paper. Therefore, we conduct an experiment using the setup described in Section 3.1 to evaluate the clustering quality under four configurations of the data-generating prior:
> - **Gaussian + Categorical**,
> - **NN-transformed + Categorical**,
> - **Gaussian + NN-transformed**,
> - **Gaussian + NN-transformed + Categorical** (our standard ZEUS model from Section 3.2).
>
> | Metric | Dataset group | Gauss. + Cat. | NN-transf. + Cat. | Gauss. + NN-transf. | Gauss. + NN-transf. + Cat. |
> |--------|---------------|---------------|-------------------|---------------------|----------------------------|
> | ARI    | Real          | 40.59        | 50.90            | 51.99              | 57.43                     |
> | ARI    | Syn. Gauss.   | 92.61        | 89.90            | 75.25              | 89.03                     |
> | ARI    | Syn. Transf.  | 73.34        | 87.04            | 71.29              | 86.33                     |
> | Rank   | Real          | 3.1029        | 2.5588            | 2.4559              | 1.8824                     |
> | Rank   | Syn. Gauss.   | 1.3500        | 3.0500            | 2.8000              | 2.8000                     |
> | Rank   | Syn. Transf.  | 3.6500        | 2.0000            | 2.0750              | 2.2750                     |
>
> As expected, the “Gaussian + Categorical” model achieves top performance on the Synthetic Gaussian datasets, while the” NN-transformed + Categorical” method excels on Synthetic Transformed data collections. Notably, the latter also performs well on Gaussian sets, as its prior is built upon transformed Gaussian mixtures. In contrast, the “Gaussian + NN-transformed” model yields relatively poor average ARI scores across both synthetic benchmarks. This drop is primarily caused by a significant decline in ARI performance on categorical datasets. Nonetheless, its average rank remains consistently competitive. For the OpenML datasets, the best performance is achieved when pre-training includes all three types of priors, underscoring the complementary importance of each prior in the training process.
>
> These additions clarify the role of individual components in ZEUS and help isolate the sources of its performance. We believe they provide a more complete and transparent evaluation of the method. We appreciate the reviewer’s feedback, which helped us improve the clarity and depth of the empirical study, and we hope these extended experiments support a more favorable assessment.

---

> > ### Comment · Reviewer_2ngo · 2025-08-05
> > **Reviewer response**
> >
> > I thank the authors for their response.
> >
> > * On the subject of ZEUS' performance: I understand the points raised py the authors. While I remain concerned about the degree to which the proposed method offers significant gains (it is very close to just applying K-Means for example), I recognize that it has advantages (e.g. calibration) that could be said to outweight its disadvantages (e.g. inference time for large datasets over naive K-Means).
> >
> > * On the subject of comparisons and model sizes: Again, I recognize the authors' points and they are indeed right that optimizing previous methods is beyond the scope of their work, however the point stands that their use of a bigger and more advanced architecture makes it difficult to disentangle this from the methodological contributions they make. That said, their point that ZEUS is trained on much more data is well taken.
> >
> > * On alternative methods & ablations: I am satisfied with the authors' responses.
> >
> > Following the above, I will raise my score to accept. Μy main concerns remain the architectural differences with competing methods and the (in my opinion) not entirelly convincing gains, but I appreciate the novel approach and the contribution regarding the training data mix and their impact.

---

> > > ### Author Response · Authors · 2025-08-05
> > >
> > > We thank the Reviewer for carefully reading our response and considering the increase in our score.
> > >
> > > We would like to clarify that it is not feasible to apply the same — or even a similar — architecture across all models. ZEUS must be implemented as a transformer, which processes the entire dataset simultaneously, whereas competing methods such as IDEC rely on autoencoders that introduce a bottleneck to identify lower-dimensional manifolds.
> > >
> > > To ensure a fair comparison, we conducted a comprehensive hyperparameter search for most methods (including latent dimensions and autoencoder size) and followed the recommendations provided in the authors’ repositories. Despite selecting the optimal configurations for these methods, we observed significant sensitivity of their results to hyperparameter choices. This further highlights the strength of our approach: ZEUS provides users with a ready-to-use, pre-trained model that requires no additional fine-tuning.
> > >
> > > As the reviewer kindly noted, ZEUS offers several additional advantages over classical baselines, including improved calibration, enhanced performance on discrete data (see our response to Reviewer 3Vjr), and detection of clusters with more sophisticated shapes (see our response to Reviewer gk7v with two moons dataset). Moreover, by releasing the code for pre-training, we aim to facilitate further development of ZEUS and encourage broader adoption of our model within the research community.
> > >
> > > We once again thank the reviewer for their positive feedback.

---

### Official Review · Reviewer_p4m9 · 2025-07-15

**Clarity:** 3
**Significance:** 3
**Originality:** 2
**Rating:** 2
**Confidence:** 3

**Summary:**

The paper suggests ZEUS as a paradigm to pre-train a network for tabular data to allow for general purpose transformations into an embedding space, in which tabular representations are more separated and thus easier to cluster. The chosen pre-training method is extensively explained and theoretically justified and performance benefits are underlined with experimental evaluation.

**Questions:**

What dimensional output does Zeus have and what role does the dimensionality play?

How exactly does the nature of zero-shot learning come into play here and how does it differ from more general transfer learning through feature extraction settings? The reference experiments seem to employ the pre-trained ZEUS network to transform features from the raw data domain to a different embedding space, in which the clustering can be performed, which seems to be a common technique under the umbrella term of transfer learning.

As ZEUS effectively only transforms the features, why weren't other approaches than k-means employed for the clustering to allow a better understanding of how valuable the transformed ZEUS representations are?

**Ethical Concerns:**

["NO or VERY MINOR ethics concerns only"]

**Limitations:**

The paper lacks some experimental validation with respect to its data basis and model architectures. Further, some details cannot be extracted due to a missing appendix.

**Paper Formatting Concerns:**

None.

**Quality:**

2

**Strengths And Weaknesses:**

Strengths:
The paper is easy to follow and presents a clear motivation and context for the suggested approach. Furthermore, the experimental evaluation shows overall a clear performance advantage of the ZEUS representations over other classification approaches.

Weaknesses:

It is not really clear how the pre-training exactly affects the model performance. It seems that ZEUS is the only model that is trained on data other than the downstream task, even though similar pre-training may also be conceivable for some of the baseline models. A comment or even additional experiments on that end may be beneficial.

Furthermore, it would be interesting to see how pre-training on real datasets would affect the ZEUS representations. It also doesn't really become clear how many samples or epochs the model has been trained on. The paper could clearly benefit from additional details and discussion of the data selection.

At many instances the paper refers to appendices for more details, which are, however, not present in the paper. An evaluation of these crucial aspects is thus not possible in the presented version, which hinders several aspects of understanding and reproducibility.

Regarding the structure, it is not clear why the related work, introducing all relevant baseline models for comparisons is introduced so late and even abbreviations are only introduced at this point.

The role of the underlying transformer model is not very clear and how its importance measures against that of the data basis. The paper would benefit from additional experiments in this regard.

minor comments.
there are some typos that need to be addressed.
l. 107: --> needs to be performed.
l. 123:  the introducing --> the introduction of

---

> ### Author Rebuttal · Authors · 2025-07-30
>
> We thank the reviewer for the thoughtful and detailed feedback, as well as for recognizing the core idea and potential of ZEUS — particularly its ability to generalize to real-world clustering tasks without downstream supervision. We appreciate the acknowledgement of the zero-shot setting and the novel use of synthetic generative processes as a meaningful contribution.
>
> We address your main concerns below:
>
> > At many instances the paper refers to appendices for more details, which are, however, not present in the paper. An evaluation of these crucial aspects is thus not possible in the presented version, which hinders several aspects of understanding and reproducibility.
>
> The appendix to our publication is available in the “Supplementary Material” section on the Openreview website, under the abstract.
>
> > It is not really clear how the pre-training exactly affects the model performance. It seems that ZEUS is the only model that is trained on data other than the downstream task, even though similar pre-training may also be conceivable for some of the baseline models. A comment or even additional experiments on that end may be beneficial.
>
> We acknowledge the reviewer’s suggestion to analyze the impact of pretraining. However, none of the baseline methods are designed to support pretraining or cross-dataset transfer. We followed the official repositories and authors’ recommendations, which do not include such procedures. Extending these methods beyond their intended scope is outside the remit of this paper.
> The performance of ZEUS variants, pre-trained under the same settings as in Section 3.1 of the main paper, except for the use of different combinations of data-generating prior (Gaussian, Categorical, and NN-transformed) is presented in the following table. We evaluate four configurations:
> - **Gaussian + Categorical**,
> - **NN-transformed + Categorical**,
> - **Gaussian + NN-transformed**,
> - **Gaussian + NN-transformed + Categorical** (our standard ZEUS model from Section 3.2).
>
> Details of these data-generating probabilistic models are provided in Appendix B.
>
> | Metric | Dataset group | Gauss. + Cat. | NN-transf. + Cat. | Gauss. + NN-transf. | Gauss. + NN-transf. + Cat. |
> |-|-|-|-|-|-|
> |ARI|Real|40.59|50.90|51.99|57.43|
> |ARI|Syn. Gauss.| 92.61|89.90|75.25|89.03|
> |ARI|Syn. Transf.| 73.34|87.04|71.29|86.33|
> |Rank|Real |3.1029|2.5588|2.4559|1.8824|
> |Rank|Syn. Gauss.|1.3500| 3.0500| 2.8000|2.8000|
> |Rank|Syn. Transf.|3.6500| 2.0000| 2.0750|2.2750|
>
> > Furthermore, it would be interesting to see how pre-training on real datasets would affect the ZEUS representations. It also doesn't really become clear how many samples or epochs the model has been trained on. The paper could clearly benefit from additional details and discussion of the data selection.
>
> ZEUS, like in-context learning methods, requires hundreds of thousands or even millions of datasets during pre-training in order to generalize effectively and perform well on entirely new downstream tasks. Unfortunately, it is not feasible to obtain such a large number of real-world datasets with well-defined and consistently labeled clusters. Pre-training on only a handful of real datasets is insufficient, and under such conditions, ZEUS would perform poorly. To address this limitation, we rely on pre-training with synthetically generated datasets.
>
> As evidence, we conduct an experiment where we split a set of real/OpenML datasets into two groups, each containing 17 datasets. The first group is used for pre-training, while the second one serves as the evaluation set. The results are presented below:
>
> | Metric | Dataset group| ZEUS | ZEUS - OpenML pre-trained |
> |-|-|-|-|
> |ARI|Real (test subset)|58.21|25.28|
> |Rank|Real (test subset)|1.0441|1.9559|
>
> Regarding the pretraining duration, ZEUS was trained for 500 epochs, each comprising 1,000 batches of single dataset sample, whereas details on the synthetic data generation process are discussed in Appendix B.
>
> > The role of the underlying transformer model is not very clear and how its importance measures against that of the data basis. The paper would benefit from additional experiments in this regard.
>
> The transformer backbone is used in ZEUS because it enables processing the entire dataset jointly — a requirement of our formulation. This design follows similar precedents in NLP and tabular learning (e.g., TabPFN). At the time of writing, transformers are the de facto state-of-the-art architecture for such use cases, and alternative architectures are either not suitable or not competitive.
>
> > What dimensional output does Zeus have and what role does the dimensionality play?
>
> The output dimensionality of ZEUS is 512. The table below summarizes our model’s clustering performance (ARI) for different output sizes: 32, 128, 512, and 768.
>
> | Metric | Dataset group | ZEUS-32 | ZEUS-128 | ZEUS-512 | ZEUS-768 |
> |-|-|-|-|-|-|
> |ARI|Real |44.10|47.81|57.43|50.79|
> |ARI|Syn. Gauss.|69.50|83.04|89.03|91.92|
> |ARI|Syn. Transf.|67.57|81.62|86.33|87.21|
> |Rank|Real|2.8971|2.9559|1.8971|2.2500|
> |Rank|Syn. Gauss.|4.0000|3.0000|1.7750|1.2250|
> |Rank|Syn. Transf.|3.9000|2.7750|1.7000|1.6250|
>
> For synthetic data, increasing the dimensionality leads to a consistent improvement in performance, whereas for OpenML datasets, there is a drop between dimensions 512 and 768, which may be caused by overfitting to the prior.
>
> > How exactly does the nature of zero-shot learning come into play here and how does it differ from more general transfer learning through feature extraction settings? The reference experiments seem to employ the pre-trained ZEUS network to transform features from the raw data domain to a different embedding space, in which the clustering can be performed, which seems to be a common technique under the umbrella term of transfer learning.
>
> We respectfully disagree. While both zero-shot learning and transfer learning involve reusing knowledge from a source domain, they are conceptually distinct. Transfer learning assumes access to the target dataset and typically involves fine-tuning the model on the downstream task — using labeled data (in supervised settings) or task-specific objectives (in unsupervised ones). In contrast, zero-shot learning requires the model to generalize to new tasks without any access to the target task during training or fine-tuning, and offers much faster inference. This is the regime we operate in.
> In our case, the ZEUS model is trained only on synthetic data generated from known probabilistic models. It never sees the downstream datasets during training and is applied to them as-is. We do not define or optimize any task-specific loss on the downstream clustering tasks. We explored fine-tuning using objectives inspired by DEC, but preliminary results were inconsistent and unstable - further motivating a pure zero-shot approach. In this sense, ZEUS learns a general mapping from data distributions to clustering structure, which can be applied without any adaptation - distinguishing it from conventional transfer learning pipelines.
>
>
> >  As ZEUS effectively only transforms the features, why weren't other approaches than k-means employed for the clustering to allow a better understanding of how valuable the transformed ZEUS representations are?
>
> ZEUS is inherently designed to form spherical clusters, so in theory, k-means (ZEUS-KM) should be a sufficient method for separating the learned representation. Its primary advantage lies in its speed, which is a crucial factor for us during inference. Moreover, as demonstrated in the tables below, its clustering quality (measured by ARI) is comparable to the more computationally intensive Gaussian Mixture Model (ZEUS-GMM). For reference, we also include results from a simplified variant of GMM with fixed identity covariance matrices (ZEUS-SGMM), as employed in Section 3.3 of the main paper. All experiments were performed using the same checkpoint and datasets as in Section 3.2. To enhance clarity, the second table presents an extended version of the comparison across 5 arbitrarily selected datasets per group.
>
> |Metric|Dataset group|ZEUS-KM|ZEUS-SGMM|ZEUS-GMM|
> |-|-|-|-|-|
> |ARI|Real|57.43|57.33|56.45|
> |ARI|Syn. Gauss.|89.03|88.18|89.46|
> |ARI|Syn. Transf.|86.33|85.57|87.22|
> |Rank|Real |1.9412|2.0441|2.0147|
> |Rank|Syn. Gauss.|1.9000|2.3250|1.7750|
> |Rank|Syn. Transf.|2.1250|2.3500|1.5250|
>
> |Dataset group|ID|ZEUS-KM|ZEUS-SGMM|ZEUS-GMM|
> |-|-|-|-|-|
> |Real|53|35.76|35.76|35.76|
> |Real|840|27.66|27.66|24.89|
> |Real|1462|92.28|92.28|92.28|
> |Real|1495 |7.35| 7.35|7.35|
> |Real|42261|85.15|83.49|85.15|
> |Syn. Gauss.|1|72.04|71.94|75.61|
> |Syn. Gauss.|6|97.89| 96.05|97.89|
> |Syn. Gauss.|10|87.32|87.32|87.32|
> |Syn. Gauss.|13|96.17|96.17|96.17|
> |Syn. Gauss.|18|83.01|81.69|83.01|
> |Syn. Transf.|0|87.42|87.23|87.51|
> |Syn. Transf.|3|80.98|77.36|81.33|
> |Syn. Transf.|7|89.57|88.69|90.72|
> |Syn. Transf.|14|98.49|98.49|98.49|
> |Syn. Transf.|18|79.70|79.70|80.12|
>
> Analysis of the average ARI scores indicates that ZEUS's output is well-suited to evaluated approaches, as there are no significant differences in their results. Deep clustering techniques such as DEC or IDEC could also potentialy be employed. Nevertheless, their reliance on intensive, dataset-specific training contradicts ZEUS’s objective of delivering efficient, zero-shot representations.
>
> We hope that our responses have addressed the reviewer’s concerns. In particular, we have added:
> - a detailed ablation on the impact of different pretraining distributions,
> - a new experiment comparing synthetic vs. real-data pretraining (Table: ZEUS - OpenML),
> - dimensionality sensitivity analysis (ZEUS-32 to ZEUS-768),
> - and a comparison of downstream clustering heads (k-means, SGMM, GMM) using identical checkpoints.
>
> These additions, which we will include in the final version of the paper, further clarify the design choices and robustness of ZEUS. We believe they strengthen the overall contribution and hope they support a more favorable evaluation.

---

> > ### Author Response · Authors · 2025-08-06
> >
> > Dear Reviewer,
> >
> > We are writing to kindly follow up on our rebuttal and to see if you’ve had a chance to review our response. We would greatly appreciate any further thoughts or feedback you might have.
> >
> > In particular, we would like to draw your attention to the detailed results included in the appendix (see supplementary material link), which were not found by the Reviewer at the initial stage of the review. Many of these address the concerns you raised and provide additional clarification.
> >
> > We also invite you to consider the discussions with other reviewers, who found our responses satisfactory. They acknowledged that our work presents a novel contribution—the first zero-shot embedding algorithm for tabular data trained exclusively on synthetic data. They also highlighted the clarity of our presentation and the thoroughness of our experimental evaluation.
> >
> > We sincerely hope you’ll have an opportunity to revisit the rebuttal and reconsider your assessment of our submission.
> >
> > Thank you again for your time and consideration.

---

### Note · Authors · 2025-08-12

Dear Area Chairs and Reviewers,

In the rebuttal phase, to address reviewers' concerns and further strengthen the paper, we added multiple new experiments, expanded baselines (IDC, SCARF, TabPFN-Unsupervised), conducted ablations on training priors, performed robustness analyses, and provided architectural clarifications. As a result, Reviewer 2ngo raised their score to acceptance, and Reviewer gk7v maintained their already positive score, explicitly expressing support for publication. Reviewer 3Vjr, despite retaining some reservations, acknowledged the novelty, soundness, and strong empirical results and we hope that the additional discussion we provided will ultimately convince them that our work constitutes a valuable contribution advancing the field.

The sole negative reviewer, p4m9, did not engage in the discussion at all, leaving initial criticisms unresolved despite our detailed, point-by-point answers. Notably, several of their key concerns were already addressed in the supplementary material, which they appear not to have noticed.

We would like to sincerely thank all reviewers for their hard work, constructive feedback, and engagement in the discussion. Their efforts have been invaluable in helping us improve and strengthen our contribution.

---

### Decision · Program_Chairs · 2025-09-17

**Decision:**

Accept (poster)

**Comment:**

This paper presents a method (Zeus) to pre-train a network for tabular data and use it for zero-shot embedding (unsupervised) that generalizes across datasets. The authors provide experimental results to demonstrate the efficacy of the approach against existing clustering techniques.

Overall, the problem is well motivated with practical applications and the paper introduces an interesting & novel approach to address the challenges described. The authors also did a good job at providing details & clarifications around several points raised by reviewers esp. around ablations. Some of the weaknesses of the paper stemmed from the limited complexity around some of the datasets used in evaluations (also highlighted by some reviewers) — for example, small #clusters, grounding the choice of transformations (and correspondingly data used for training) wrt learnability & generalizability of the model, extensions for the architecture choice, etc. The authors address several of the concerns in the rebuttal which def. strengthens the claims, suggest these along with other important feedback to be incorporated into the paper to make it well-rounded and standalone.